# A model of Notch signalling control of angiogenesis: Evidence of a role for Notch ligand heterodimerization

**Daipeng Chen**[1,2‡], **Xinxin Liu**[3‡], **Haijiang Wang**[3,4‡], **Roeland M.H. Merks**[2,5], **David A. Baker**[3*]

**1** School of Mathematics and Statistics, Xi'an Jiaotong University, Xi'an, China, **2** Mathematical Institute, Leiden University, Leiden, The Netherlands, **3** Department of Cell and Chemical Biology, Leiden University Medical Center (LUMC), Leiden, The Netherlands, **4** Department of General Surgery, The First Affiliated Hospital, Xi'an Jiaotong University, Xi'an, China, **5** Institute of Biology Leiden, Leiden University, Leiden, The Netherlands

‡ These authors are joint first authors.
* d.baker@lumc.nl

## Abstract

The ubiquitous Notch receptor signalling network is essential for tissue growth and maintenance. Operationally, receptor activity is regulated by two principal, counterposed mechanisms: intercellular Notch transactivation triggered by interactions between receptors and ligands expressed in neighbouring cells; intracellular cis inhibition mediated by ligands binding to receptors expressed in the same cell. Moreover, different Notch receptor/ligand combinations are known to elicit distinct molecular and cellular responses, and together, these phenomena determine the strength, the duration and the specificity of Notch receptor signalling. To date, it has been assumed that these processes involve discrete ligand homomers and not heteromeric complexes composed of more than one ligand species. In this study, we explore the molecular basis of the opposing actions of the Notch ligands, DLL4 and JAG1, which control angiogenic sprouting. Through a combination of experimental approaches and mathematical modelling, we provide evidence that two mechanisms could underpin this process: 1) DLL4 rather than JAG1 induces efficient Notch1 receptor transactivation; 2) JAG1 directly blocks DLL4-dependent cis-inhibition of Notch signalling through the formation of a JAG1/DLL4 complex. We propose a new model of Notch signalling that recapitulates the formation of tip and stalk cells, which is necessary for sprouting angiogenesis.

## Author summary

The ancient Notch signalling pathway is a central regulator of metazoan cell fate, and corruption of it has been implicated in numerous diseases including the majority of solid tumours. Since its discovery in the second decade of the last century one central question to emerge is how such a wide range of different cellular/physiological processes can be governed by such a relatively simple repertoire of Notch receptors and ligands. In this study, using a combination of experimental approaches and mathematical modelling,

**Data availability statement:** All relevant data are within the manuscript and its Supporting Information files. Code running on MATLAB R2021a can be accessed via: https://github.com/DaipengChen/Notch-Angiogenesis.

**Funding:** This work was supported by the Dutch Cancer Society (30861) to DAB, the Nederlandse Organisatie voor Wetenschappelijk Onderzoek grant NWO/ENW-VICI 865.17.004 to RMHM, the Cancer Genomics Centre Netherlands (CGC.NL) to XL. DC and HW were recipients of Chinese Scholarship Council (CSC) funding as part of the CSC Joint PhD Program on Artificial Intelligence and Bioscience between Leiden University and Xi'an Jiaotong University. The funders had no role in study design, data collection and analysis, decision to publish, or preparation of the manuscript.

**Competing interests:** The authors have declared that no competing interests exist.

we provide the first evidence that Notch ligands can directly bind to each other, and that this novel mechanism of ligand heterodimerization could help determine the strength, the duration and the specificity of Notch signalling. Our model faithfully recapitulates both our presented experimental results as well as the recently published work of others, and provides a novel perspective for understanding Notch-regulated biological processes, most notably angiogenesis.

## Introduction

The ancient and highly conserved Notch receptor signalling pathway is indispensable for embryonic and post-embryonic tissue growth and development [1,2]. It is formed of large (approximately 300 kDa) single pass cell membrane receptors, and ligands which can also be expressed at the cell surface. The overall size, structure and known function of these core receptor/ligand components has remained relatively unchanged over evolutionary time although the number of Notch receptor and ligand family members encoded by different genera varies, for example, invertebrates such as *Drosophila* possess a single type of Notch receptor controlled by two distinct ligands, whilst vertebrates express up to four distinct receptor paralogues (Notch1–4) and five different ligands: Jagged (JAG)1, JAG2, Delta-Like (DLL)1, DLL3, and DLL4 [3]. Uniquely amongst cell surface receptors, Notch itself can directly activate gene transcription as part of a DNA-bound activation complex, a signal triggered via a *trans* interaction between the extracellular domain of the receptor and Notch ligands expressed by neighbouring cells. Upon ligand binding, the resultant biomechanical strain/pulling forces cause structural changes which expose specific enzyme cleavage sites in the membrane-proximal region of the receptor [4–8]. Subsequently, a defined series of proteolytic events terminates in the γ-secretase-mediated cleavage of the Notch intracellular domain [9,10], which translocates to the nucleus whereupon it stimulates expression of Notch target genes [11,12]. An important counterbalance to Notch signalling transactivation is a regulatory mechanism termed cis-inhibition by which ligands block the activity of receptors expressed in the same cell [13–15]. Thus, in this view, the net receptor signalling output is determined by the relative 'strength' of these two counterposed processes (transactivation and cis-inhibition).

Whilst, the basic features of Notch receptor transactivation, outlined above, have been delineated in some detail, a number of fundamental issues remain enigmatic. One question relates to the precise nature and significance of cis inhibition and how it counteracts receptor transactivation. Another central question concerns how the manifold potential receptor-ligand combinations will give rise to the different signalling outputs necessary for tissue patterning. In this context, recent work has dissected, at the molecular level, the cellular responses to different ligands, which can stimulate either discrete pulses of Notch activity (in the case of DLL1) or a sustained period of signalling (in the case of DLL4) yielding distinct gene expression outcomes [16,17]. At the tissue level, a classic example of the divergent roles of different Notch ligands is found during sprouting angiogenesis wherein DLL4 was shown to constrain sprouting opposed by the pro-angiogenic properties of JAG1 [18,19]. Of significant interest, in this *in vivo* setting, was the notion that JAG1 functions by antagonizing the activity of DLL4, highlighting the possibility of crosstalk/competition between the ligands, though the specific molecular details have not been elucidated.

In this light, other potential facets of Notch signalling merit consideration. For instance, in the absence of available evidence, it has been generally assumed that cis and trans receptor/ligand interactions are essentially monomeric in nature, and involve binding of one species of ligand and a single species of receptor rather than complexes of different receptor types,

or different ligand types. There are, however, reasons to suppose that the actual picture may be more complicated. Both receptors and ligands harbour multiple EGF-like repeats, which are known to mediate protein-protein interactions [20], and we have recently shown that homomeric DLL4 ligand complexes could promote cis inhibition of Notch receptor activity [15]. Moreover, the fact that all Notch ligands share a common overall architecture raises the possibility of a potential biological role for heteromeric ligand complex formation, until now, a mechanism which has not been reported.

In this study, we combine experimental approaches and mathematical modelling to explore the molecular control of Notch1 receptor activity by the ligands, DLL4 and JAG1. Our work uncovers a previously overlooked role for heteromeric JAG1/DLL4 complexes, which could enable angiogenic sprouting by regulating the functional differentiation of endothelial cells into leading edge tip cells and trailing stalk cells.

## Results

It is established that angiogenic sprouting results from the differential regulation of Notch1 receptor activity by DLL4 and JAG1 (see [18,19]). To understand the molecular details of this mechanism we first investigated the possible biochemical interactions which could control this process.

### DLL4 but not JAG1 efficiently stimulates transactivation of Notch1

To test the relative Notch receptor transactivation potential of the ligands DLL4 and JAG1, we performed quantitative reporter assays. Fig 1A shows that whilst DLL4 (and JAG2) efficiently promoted Notch1 transactivation, JAG1 was unable to elicit detectable Notch1 transactivation. Conversely, Fig 1B shows that JAG1 could induce transactivation of Notch2 and Notch3 suggesting that the transactivation block is specific to Notch1. To elucidate a possible molecular basis of this difference we assessed binding of the ligands to Notch receptors expressed at the cell surface. Fig 1C and 1D show that DLL4, but not JAG1, bound relatively efficiently to Notch1 (Fig 1C). In contrast to its lack of binding to Notch1, JAG1 efficiently associated with both Notch2 and Notch3 (Fig 1D). Collectively, these data demonstrate that DLL4, can bind to and effectively trigger Notch1 receptor transactivation, whereas JAG1 fails to bind to the Notch1 receptor with sufficiently high affinity to elicit robust receptor transactivation.

We next explored the respective roles of DLL4 and JAG1 in Notch1 cis inhibition.

### DLL4 but not JAG1 promotes strong cis inhibition of Notch1 receptor signalling

Fig 2A shows that DLL4, as expected, potently blocked transactivation of Notch1 when both Notch1 and DLL4 are expressed in the same cell (cis inhibition; see [13]). By comparison, JAG1 could not similarly induce strong (cis) inhibition of Notch1 signalling (Fig 2A), whereas the closely related JAG2 ligand, in common with DLL4, promoted strong cis inhibition of Notch1 activity. We have previously shown that DLL4-dependent cis inhibition of Notch activity requires ligand dimerization/oligomerization (herein referred to simply as dimerization) [15]. In this light, we compared DLL4, JAG2 and JAG1 ligand dimerization. In agreement with our prior observations, both DLL4 and JAG2 could biochemically self-associate, however, JAG1 could not detectably form homodimers, which could explain the observed differences in their relative effectiveness in Notch1 receptor cis inhibition (Fig 2B). The difference between JAG1 and JAG2 was particularly surprising since they display a high degree of overall structural similarity. Indeed, the only significant difference between the two ligands is a von Willebrand factor type C (VWFC) domain located in the juxtamembrane region of the

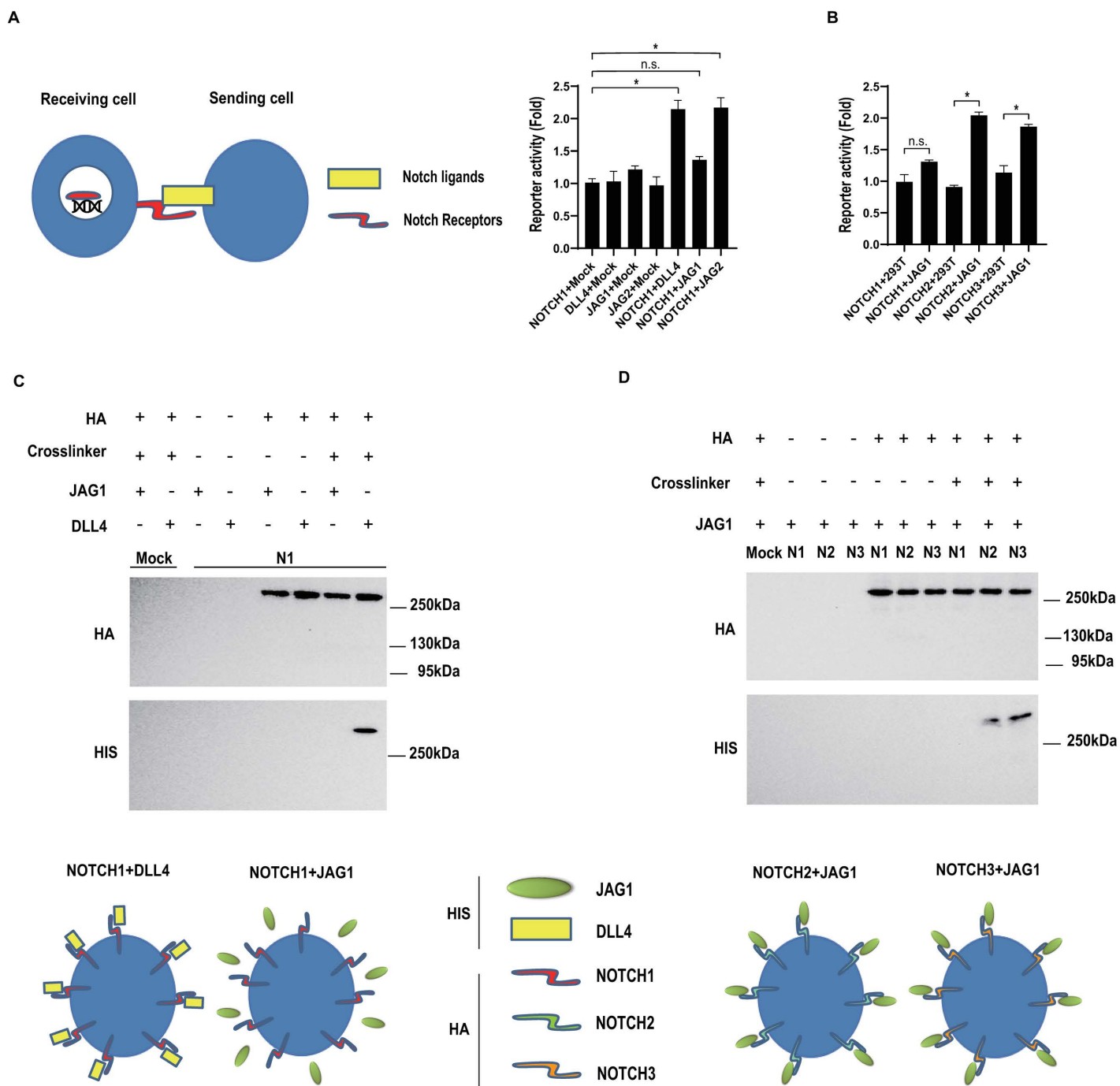

**Fig 1. DLL4, but not JAG1, efficiently stimulates Notch1 transactivation.** (A) Luciferase reporter assays were performed as described in the Methods, and depicted schematically (left panel). (B) As illustrated in (A), 293T cells co-expressing a Notch activity luciferase reporter together with the indicated Notch receptor or ligand constructs (Receiving cells) were co-cultured with cells stably expressing either DLL4 or JAG1 (sender cells) to enable transactivation. For each analysis, reporter activity was normalized using Renilla luciferase. Protein expression levels were determined by Western blotting of cell lysates. Each condition in each experiment was performed in triplicate and error bars represent the standard deviation of the mean. Experiments were performed three times. Representative experiments are shown. (C) Cell surface ligand-receptor binding assays were performed as described in the Methods, and depicted schematically. Cells stably expressing HA epitope tagged Notch1 were incubated with or without the indicated HIS epitope tagged ligand. Complexes were resolved by immunoprecipitation using an anti-HA antibody. Total receptor protein levels and ligand-receptor binding was determined by Western blotting using the indicated antibodies (anti-HA for the receptors, anti-HIS for the ligands). (D) To compare the relative levels of JAG1 binding to either Notch1, Notch2 or Notch3, cells stably expressing (separately) HA epitope tagged versions of each Notch paralogue, were incubated with HIS epitope tagged JAG1. Complexes were resolved by immunoprecipitation using an anti-HA antibody. Total receptor protein levels and ligand-receptor binding was determined by Western blotting using the indicated antibodies (anti-HA for the receptors, anti-HIS for the ligands).

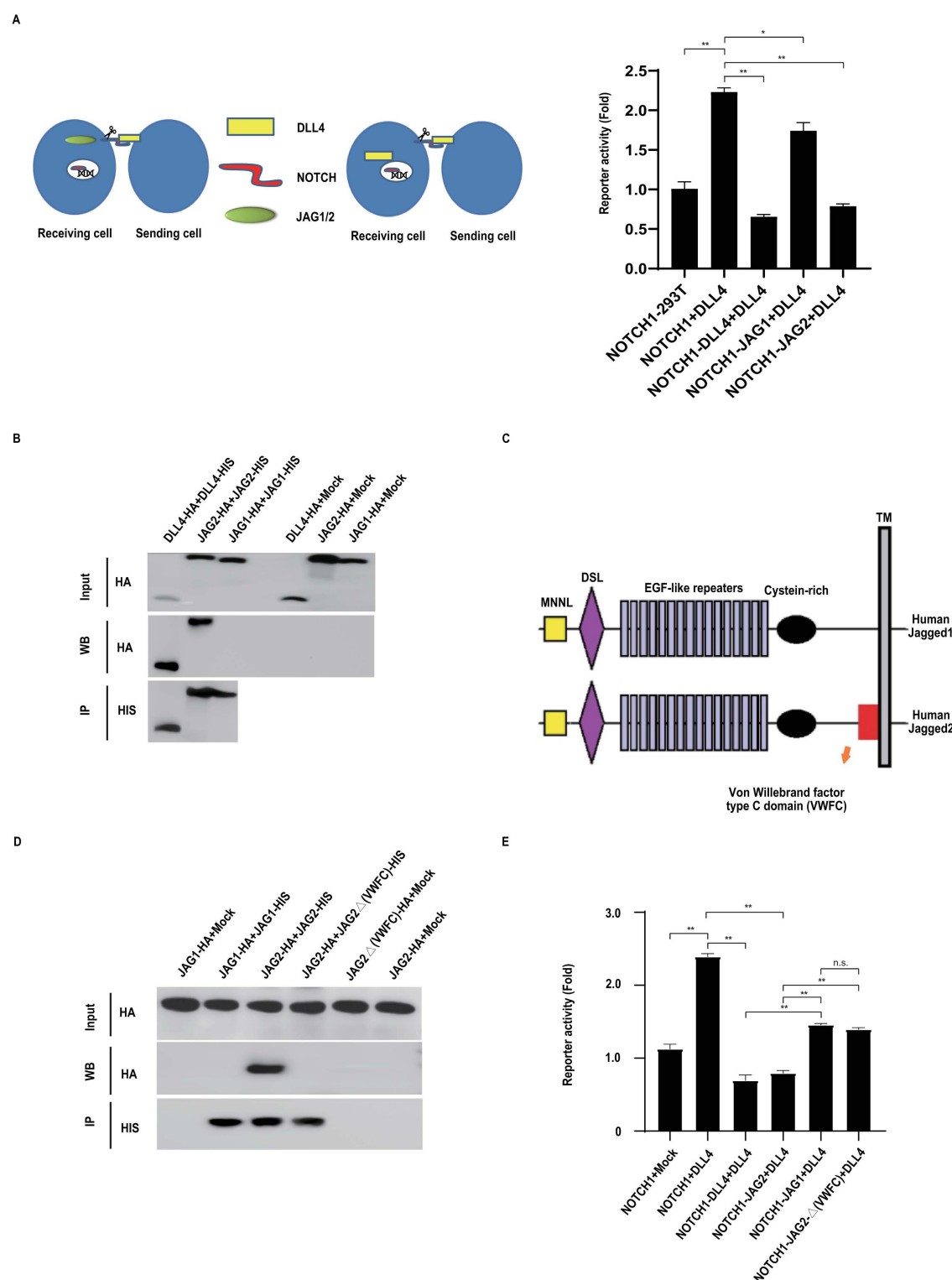

**Fig 2. DLL4 (and JAG2), but not JAG1, efficiently mediates Notch1 receptor cis inhibition.** (A) Luciferase reporter assays were performed as described in the Methods, and depicted schematically (left panel). 293T cells co-expressing a Notch activity luciferase reporter together with the indicated Notch receptor or ligand constructs (Receiving cells) were co-cultured with cells stably expressing DLL4 (sender cells) to enable transactivation. For each analysis, reporter activity was normalized using Renilla luciferase. Protein expression levels were determined by Western blotting of cell lysates (see S9 Fig). Each condition in each experiment was performed

in triplicate and error bars represent the standard deviation of the mean. Experiments were performed three times. Representative experiments are shown. (B) DLL4 (and JAG2), but not JAG1, can efficiently self-associate. The indicated combinations of HA and HIS epitope tagged versions of either DLL4, JAG1 or JAG2 were transfected into 293T cells. Ligand complexes were resolved by immunoprecipitation using Nickel-NTA agarose (which binds $HIS_6$ epitope-tagged ligands) and visualized by Western blotting with the shown antibodies. (C) Schematic representation of the JAG1 and JAG2 ligands highlighting the von Willebrand factor type C (VWFC) domain. (D) The VWFC domain mediates JAG2 self-association. As in (B), the indicated combinations of HA and HIS epitope tagged versions of either JAG1 or JAG2 were transfected into 293T cells. The JAG2-Δ(VWFC) construct lacks the VWFC domain. Ligand complexes were resolved by immunoprecipitation using Nickel-NTA agarose (which binds $HIS_6$ epitope-tagged ligands) and visualized by Western blotting with the shown antibodies. (E) The VWFC is necessary for efficient JAG2-depedent cis inhibition of Notch1. Luciferase reporter assays were performed as described in (A).

JAG2 extracellular domain (Fig 2C). Two lines of evidence support the idea that homodimer formation by DLL4 and JAG2 is a prerequisite for full Notch1 cis inhibition, and that JAG1 cannot induce full Notch1 cis inhibition because it cannot form homodimers. First, deletion of the JAG2 VWFC domain blocked homodimer formation (Fig 2D). This is consistent with previous studies which have shown, for different proteins, that the VWFC domain can mediate protein oligomerization [21]. Thus, in the absence of the VWFC domain, with respect to ligand self-association, JAG2 exhibits biochemical characteristics comparable to JAG1. Second, deletion of the JAG2 VWFC domain abrogated JAG2-dependent cis inhibition of Notch1 to a degree comparable to JAG1 (Fig 2E). It is notable that in contrast to the relatively weak trans Notch1-JAG1 binding, JAG1 can bind relatively efficiently to Notch1 in cis (see S1 Fig), consistent with the view that the lack of JAG1-mediated cis inhibition of Notch1 does not result from a lack of cis receptor-binding, but is due to the fact that JAG1 cannot form homodimers, which is necessary for cis inhibition (see [15]).

In sum, these data suggest that JAG1, uniquely for Notch ligands, fails to form homodimers/oligomers resulting in an inability to effect full cis inhibition of Notch1 receptor signalling.

## JAG1 directly blocks DLL4-mediated cis inhibition of Notch1 receptor signalling via the formation of a heteromeric JAG1/DLL4 complex

A seminal study revealed that JAG1 antagonism of the function of DLL4 determines the balance between tip cells and stalk cells necessary for normal capillary sprouting during mouse retina blood vessel development [18], although it is not yet clear if this is via direct/indirect means or at which level of Notch signalling JAG1 acts (receptor transactivation, receptor cis inhibition or both). It is known that the tip cells of the angiogenic front of the growing plexus of the mouse retina express high levels of DLL4 (and lack JAG1 expression) whilst the neighbouring stalk cells express both DLL4 and JAG1 [18]. Accordingly, to investigate the possible biochemical interactions, which might be involved in this process, we first tested Notch1 receptor activity in cells expressing both JAG1 and DLL4, in response to DLL4 expressed on neighbouring cells (see Fig 3A). Fig 3B shows that alone, as expected (see Fig 2), DLL4, but not JAG1, efficiently blocked Notch1 receptor transactivation (cis inhibition). Strikingly, when DLL4 and JAG1 are co-expressed, there is a near-complete release of the block to receptor transactivation (Fig 3B), whereas co-expression of JAG2 had no measurable impact on DLL4-mediated cis inhibition, suggesting this effect is strictly JAG1-dependent (see Fig 3B). To determine if this effect is indirect or instead results from a direct interaction between the JAG1 and DLL4 ligands, we assessed if the ligands could (biochemically) associate. Fig 3C and 3D show that DLL4 and JAG1 can form a complex (Fig 3C) and that EGF-like repeat 7 (in the proximity of the transmembrane domain) of DLL4 is necessary for this interaction (Fig 3D). We surmised that if direct binding of JAG1 to DLL4 is necessary to block cis inhibition

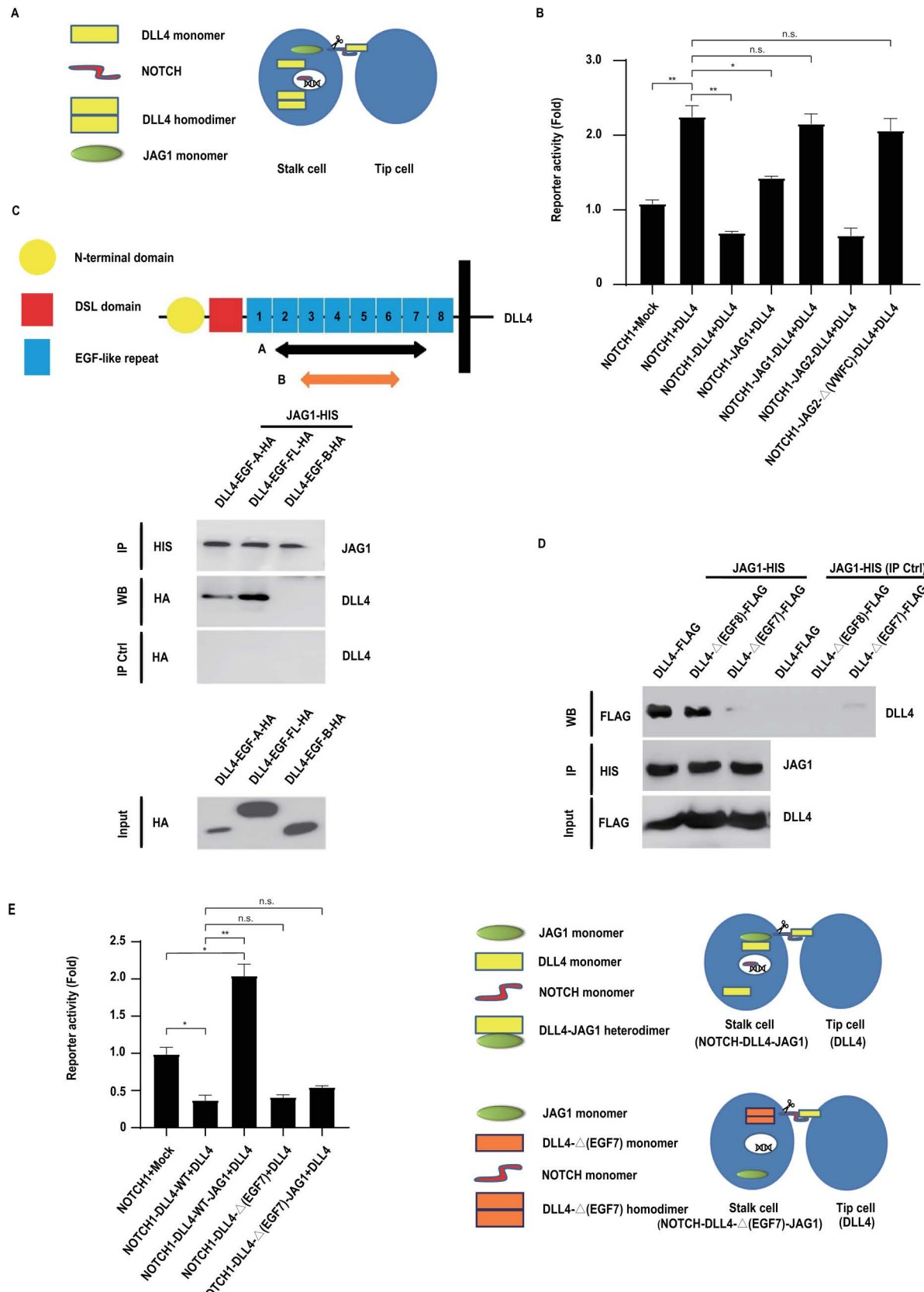

**Fig 3. A DLL4-JAG1 heterodimer blocks DLL4-mediated Notch1 receptor cis inhibition and enables efficient Notch1 receptor trans-activation.** (A) Luciferase reporter assays were performed as described in the Methods, and depicted schematically (left panel). (B) 293T

cells co-expressing a Notch activity luciferase reporter together with the indicated Notch receptor and ligand constructs (to represent stalk cells) were co-cultured with cells stably expressing DLL4 (to represent tip cells) to enable receptor transactivation. The JAG2-Δ(VWFC) construct lacks the VWFC domain. For each analysis, reporter activity was normalized using Renilla luciferase. Protein expression levels were determined by Western blotting of cell lysates. Each condition in each experiment was performed in triplicate and error bars represent the standard deviation of the mean. Experiments were performed three times. Representative experiments are shown. (C) Biochemical mapping of the DLL4-JAG1 heterodimerization motif. The indicated epitope tagged constructs were transfected into 293T cells (a schematic representation of the constructs is shown - left panel). Ligand complexes were resolved by immunoprecipitation using Nickel-NTA agarose (which binds HIS$_6$ epitope-tagged ligands) and visualized by Western blotting with the shown antibodies. (D) The DLL4 EGF-like repeat 7 (see C), is necessary for DLL4-JAG1 heterodimerization. The indicated epitope tagged constructs were transfected into 293T cells. DLL4-(EGF7) and DLL4-(EGF8) lack the EGF-like repeat 7 and the EGF-like repeat 8 (see C), respectively. Ligand complexes were resolved by immunoprecipitation using Nickel-NTA agarose (which binds HIS$_6$ epitope-tagged ligands) and visualized by Western blotting with the shown antibodies. (E) Cis DLL4-JAG1 heterodimerization enables Notch1 receptor transactivation. Luciferase reporter assays were performed as described in (B) and are displayed schematically (right panel).

to enable efficient Notch1 transactivation, a DLL4 mutant lacking EGF-like repeat 7 (which cannot form a complex with JAG1, see Fig 3D), would be resistant to the effects of JAG1 expressed in the same cell and execute full cis inhibition of Notch1 signalling. Fig 3E (and S2 Fig) show that this is the case, that is, the level of Notch1 receptor cis inhibition in the presence of a combination of mutant DLL4 (lacking EGF-like 7) and JAG1 is similar to the level of cis inhibition mediated by DLL4 alone, resulting in a block to DLL4-mediated Notch cleavage (S2 Fig) and transactivation (Fig 3E) in these cells.

These data argue that a heteromeric JAG1/DLL4 complex, formed when the ligands are expressed in the same cell, relieves Notch1 receptor cis inhibition and thereby enables efficient Notch1 receptor signalling.

Overall, our experimental work raises the possibility that two factors could underlie the mechanism by which JAG1 antagonizes DLL4 to enable appropriate tip and stalk cell selection to drive angiogenic sprouting. One, in agreement with prior work [16], DLL4, but not JAG1, can bind to Notch1 (in trans) with sufficiently high affinity to stimulate receptor activation. Two, JAG1 directly blocks DLL4-mediated cis inhibition, via the formation of a heteromeric DLL4/JAG1 complex, which allows transactivation of Notch receptors expressed in the same cell.

## Modelling DLL4/JAG1-Notch signalling

To explore the potential biological implications of our biochemical findings presented above, we extended our earlier theoretical model of DLL4-Notch signalling [15] to produce a new model which takes account of the observed differences in affinity of the Notch1 receptor for (trans) DLL4 and (trans) JAG1, as well as heteromeric DLL4/JAG1 complex formation. In any given cell, the new model is presented mathematically as Eq. (1),

$$
\begin{aligned}
\frac{dD}{dt} &= b_D - \beta D - 2k_{ho}D^2 - k_{he}JD - k_t N_{ext}D, \\
\frac{dJ}{dt} &= b_j - \beta J - k_{he}DJ - \alpha k_t N_{ext}J, \\
\frac{dC}{dt} &= k_{ho}D^2 - \beta C - k_{ci}NC, \\
\frac{dN}{dt} &= b_N - \beta N - k_{ci}CN - k_t D_{ext}N - \alpha k_t J_{ext}N, \\
\frac{dS}{dt} &= k_t D_{ext}N + \alpha k_t J_{ext}N - \beta_S N,
\end{aligned}
\tag{1}
$$

where $b_D$, $b_J$ and $b_N$ denote the production rates of DLL4 ($D$), JAG1 ($J$) and Notch1 receptor ($N$), respectively. The variable $C$ indicates the amount of DLL4 homodimers,

and $k_{ho}$ represents the rate of DLL4-DLL4 dimerization. The heterodimer of DLL4 and JAG1 is an inactive complex, and $k_{he}$ represents the rate of DLL4-JAG1 dimerization. The proteins described in the model are assumed to be degraded at a constant rate given by $\beta$ and $\beta_S$, which define the degradation rates of Notch1 ligands and receptors, and the free Notch1 intracellular domain ($S$), respectively. In our model (1), Notch1 is trans-activated by DLL4 (at rate $k_t$) and JAG1 (at rate $\alpha k_t$) expressed in neighbouring cells, denoted by $D_{ext}$ (DLL4) and $J_{ext}$ (JAG1), and is cis-inhibited by DLL4 homodimers (at rate $k_{ci}$) co-expressed in the same cell. A comprehensive description of all parameters is detailed in Table 1.

Model (1) describes possible ligand and receptor interactions within the same cell and between cells. In common with previous publications [14,15,17,28,29], here we assumed that the functional protein-protein interactions proceed irreversibly. A novel component of this model is that it considers the experimentally observed DLL4-JAG1 heterodimerization and the different affinities for Notch1 of trans DLL4 and JAG1. Parameter $\alpha$ sets the ratio of Notch1-JAG1 affinity to Notch1-DLL4 affinity. Specifically, $\alpha = 0$ means that JAG1 does not mediate the trans-activation of Notch1, whereas $\alpha = 1$ means that JAG1 and DLL4 transactivate Notch1 with equal affinity. Consistent with the experimental findings, DLL4 homodimers, but DLL4-JAG1 heterodimers, mediate Notch1 receptor cis-inhibition.

Table 1. Description and baseline values of parameters used in simulations.

| Par | Description | Values | Units | Source |
|---|---|---|---|---|
| $b_D$ | Baseline production rate of DLL4 | 100 | molec / hour | According to [53] |
| $b_J$ | Baseline production rate of JAG1 | 100 | molec / hour | According to [53] |
| $b_N$ | Baseline production rate of Notch receptor | 500 | molec / hour | According to [53] |
| $b_R$ | Baseline production rate of VEGFR2 | 800 | molec / hour | According to [53] |
| $\beta$ | Degradation rate of typical proteins | 0.1 | $hour^{-1}$ | Estimated from [54] |
| $\beta_S$ | Degradation rate of free Notch Intracellular Domain and free VEGFR2 Intracellular Domain | 0.5 | $hour^{-1}$ | Estimated from [55] |
| $k_{ho}$ | Homodimerization rate of DLL4 | $1*10^{-4}$ | $molec^{-1}*hour^{-1}$ | [15] |
| $k_{he}$ | Heterodimerization rate of DLL4 and JAG1 | $1*10^{-3}$ | $molec^{-1}*hour^{-1}$ | Assumed |
| $k_t$ | Trans-activation rate | $5*10^{-5}$ | $molec^{-1}*hour^{-1}$ | [15,28,29] |
| $k_{ci}$ | Cis-inhibition rate | $6*10^{-4}$ | $molec^{-1}*hour^{-1}$ | [15] |
| $k_v$ | VEGF-VEGFR2 binding rate | $5*10^{-5}$ | $molec^{-1}*hour^{-1}$ | [28,29] |
| $\lambda D$ | Involved in regulating DLL4 production | 10 | dimensionless | Estimated from [23] |
| $\lambda_R$ | Involved in regulating VEGFR2 production | 0 | dimensionless | [28,29,41] |
| $\theta_D$ | Involved in regulating DLL4 production | 200 | dimensionless | [28,29,41] |
| $\theta_R$ | Involved in regulating VEGFR2 production | 30 | dimensionless | [28,29,41] |
| $n_D$ | Involved in regulating DLL4 production | 2 | dimensionless | [28,29,41] |
| $n_R$ | Involved in regulating VEGFR2 production | 2 | dimensionless | [28,29,41] |
| $\alpha$ | The ratio of Notch1-JAG1 affinity to Notch1-DLL4 affinity in trans | $1*10^{-4}$ | dimensionless | Assumed |
| $V_{ext}$ | Concentrations of free VEGF at the growing front of angiogenic sprouting | 2500 | molec | Assumed |

## Modelling DLL4/JAG1-Notch-VEGF signalling

Since angiogenic sprouting involves a tightly regulated interplay between Notch signalling and vascular endothelial growth factor (VEGF) receptor (VEGFR) signalling, based upon our new experimental results, we further elaborated our DLL4/JAG1-Notch signalling model (1) and propose a new theoretical model of DLL4/JAG1-Notch-VEGF signalling. In this model, free VEGF binds to VEGFR2 and stimulates intracellular VEGFR2 signalling, which upregulates the production of DLL4 in the same cell, as previously reported [22,23]. In turn, DLL4 stimulates Notch1 trans-activation and the intracellular Notch1 signalling downregulates the production of VEGFR2, a feedback mechanism which has been established experimentally [24,25]. With the addition of VEGFR2 signalling, the new model is presented mathematically as,

$$
\begin{cases}
\dfrac{dD}{dt} = H(V;\theta_D,\lambda_D,n_D)b_D - \beta D - 2k_{ho}D^2 - k_{he}JD - k_t N_{ext}D, \\[2mm]
\dfrac{dJ}{dt} = b_J - \beta J - k_{he}DJ - \alpha k_t N_{ext}J, \\[2mm]
\dfrac{dC}{dt} = k_{ho}D^2 - \beta C - k_{ci}NC, \\[2mm]
\dfrac{dN}{dt} = b_N - \beta N - k_{ci}CN - k_t D_{ext}N - \alpha k_t J_{ext}N, \\[2mm]
\dfrac{dS}{dt} = k_t D_{ext}N + \alpha k_t J_{ext}N - \beta_S S, \\[2mm]
\dfrac{dR}{dt} = H(S;\theta_R,\lambda_R,n_R)b_R - \beta R - k_v V_{ext}R, \\[2mm]
\dfrac{dV}{dt} = k_v V_{ext}R - \beta_S V,
\end{cases}
\tag{2}
$$

where $b_R$ denotes the baseline production rate of VEGFR2 ($R$), and the free VEGF ($V_{ext}$) induces intracellular VEGFR2 signalling ($V$) via binding to VEGFR2 (at rate $k_v$). Other variables and parameters in model (2) represent the same biological terms described in model (1) and Table 1. The regulation of DLL4 production by VEGFR2 signalling and the regulation of VEGFR2 production by Notch signalling is given by a Hill function:

$$
H(X;\theta,\lambda,n) = \frac{\lambda X^n + \theta^n}{X^n + \theta^n},
$$

where the parameters $\theta$, $\lambda$ and $n$ control the shape of this function. Specifically, $\lambda > 1$ means upregulation and $\lambda < 1$ means downregulation. In the absence of VEGFR2 signalling (i.e., $b_R = 0$), Eq. 2 reduces to Eq. 1.

## Exploring the opposing effects of DLL4 and JAG1 on Notch signalling

We first investigated the effects of DLL4 and JAG1 on Notch1 signalling by analyzing model (1) theoretically. Let ($\bar{D}$, $\bar{J}$, $\bar{C}$, $\bar{N}$, $\bar{S}$) be the positive steady state of model (1). According to the first two equations in the DLL4/JAG1-Notch signalling model (1), the steady-state levels of DLL4 and JAG1 satisfy the following equations:

$$
\bar{D}^3 + (A_1 + B_1)\bar{D}^2 + \left(\frac{b_J - b_D}{2k_{ho}} + A_1 B_1\right)\bar{D} - \frac{b_D}{2k_{ho}}B_1 = 0,
\tag{3}
$$

$$\bar{J}^3 + \left( \frac{b_D - b_J}{\beta + \alpha k_t N_{ext}} + A_2 - \left( \beta + \alpha k_t N_{ext} \right) B_2 \right) \bar{J}^2 + \left( 2 b_J B_2 - \frac{b_J A_2}{\beta + \alpha k_t N_{ext}} \right) \bar{J} - \frac{b_J^2 B_2}{\beta + \alpha k_t N_{ext}} = 0, \quad (4)$$

where

$$A_1 = \frac{\beta + k_t N_{ext}}{2 k_{ho}}, \quad B_1 = \frac{\beta + \alpha k_t N_{ext}}{k_{he}}, \quad A_2 = \frac{\beta + k_t N_{ext}}{k_{he}}, \quad B_2 = \frac{2 k_{ho}}{k_{he}^2}.$$

Let

$$F\left(D; b_D, b_J\right) = D^3 + \left(A_1 + B_1\right) D^2 + \left( \frac{b_J - b_D}{2 k_{ho}} + A_1 B_1 \right) D - \frac{b_D}{2 k_{ho}} B_1.$$

$F\left(\bar{D}; b_D, b_J\right) = 0$, $F_D\left(\bar{D}; b_D, b_J\right) > 0$, thus the implicit function theorem implies that there is a smooth function $h : R^2 \to R$ such that $\bar{D} = h\left(b_D, b_J\right)$ and

$$\frac{\partial h}{\partial b_D} = -\frac{F_{b_D}}{F_D} = \frac{D}{2 k_{ho} F_D} > 0, \quad \frac{\partial h}{\partial b_J} = -\frac{F_{b_J}}{F_D} = -\frac{D}{2 k_{ho} F_D} < 0 \quad (5)$$

Inequalities (5) show that $\bar{D}$ is an increasing function of $b_D$ and a decreasing function of $b_J$. Similarly, $\bar{J}$ is a decreasing function of $b_D$ and an increasing function of $b_J$ by analyzing Eq. (4). This means that an increasing production rate of JAG1 reduces the amount of available free DLL4 in the same cell, and an increasing production rate of DLL4 reduces the amount of available free JAG1 in the same cell.

In particular, if the parameter $k_{he} \gg \max\left\{ \beta + k_t N_{ext}, \sqrt{k_{ho}} \right\}$ (i.e., the heterodimerization of DLL4 and JAG1 is very strong), Eqs. (3 and 4) can be approximated as:

$$\bar{D}^3 + \frac{\beta + k_t N_{ext}}{2 k_{ho}} \bar{D}^2 + \frac{b_J - b_D}{2 k_{ho}} \bar{D} = 0,$$
$$\bar{J}^3 + \frac{b_D - b_J}{\beta + \alpha k_t N_{ext}} \bar{J}^2 = 0. \quad (6)$$

If $b_D > b_J$ (i.e., the production rate of DLL4 becomes larger than the production rate of JAG1 in the cell), this leads to:

$$\bar{D} = \sqrt{\left( \frac{\beta + k_t N_{ext}}{4 k_{ho}} \right)^2 + \frac{b_D - b_J}{2 k_{ho}}} - \frac{\beta + k_t N_{ext}}{4 k_{ho}}, \quad \bar{J} = 0. \quad (7)$$

Conversely, if $b_J > b_D$ (i.e., the production rate of JAG1 is greater than the production rate of DLL4 in the cell), this leads to:

$$\bar{D} = 0, \quad \bar{J} = \frac{b_J - b_D}{\beta + \alpha k_t N_{ext}}. \quad (8)$$

Thus, Eqs. (7 and 8) predict that, in the context of Notch1 receptor activity, the heterodimerization of DLL4 and JAG1 induces a mutual inactivation of the two ligands in the same cell (see S3 Fig). Which ligand predominates to control Notch signalling depends on their relative production rates in the cell.

We next tested if numerical simulations of the DLL4/JAG1-Notch signalling model (1) could recapitulate the observed opposing effects of DLL4 and JAG1 on Notch1 signalling (see Figs 2 and 3), and previously published data (see Fig 6 in [18]) similarly describing the

different roles of DLL4 and JAG1 in the control of Notch1 signalling. We first considered the effects of DLL4 and JAG1 on Notch receptor cis-inhibition. Fig 4A schematically depicts the simulation setup where the level of DLL4 expressed at the cell surface of a sending cell is fixed and we simulate the dynamics of Notch, DLL4 and JAG1 in the receiving cell. This arrangement reproduces our experimental conditions (see Figs 2 and 3), and the pattern of ligand expression found in the sprouting front of the developing mouse retina [18]. In Fig 4B, we simulated the model to (near) steady-state levels of Notch activity (free Notch Intracellular domain) driven by a balance between the production rate of cis-DLL4 in the receiving cell (expressing Notch receptor), which is inhibitory, and fixed levels of trans-DLL4 in a sending cell, which is stimulatory. Increasing the production rate of cis-DLL4 resulted in an inhibition of Notch receptor activity (Fig 4B), which is consistent with our experimental data (see Fig 2). Next, in Fig 4C, we tested the effect of JAG1 on Notch receptor activity when the production rate of cis-DLL4 in the receiving cell is sufficient to inhibit Notch receptor activity. Clearly, model (1), which incorporates DLL4-JAG1 heterodimerization, but not a model lacking

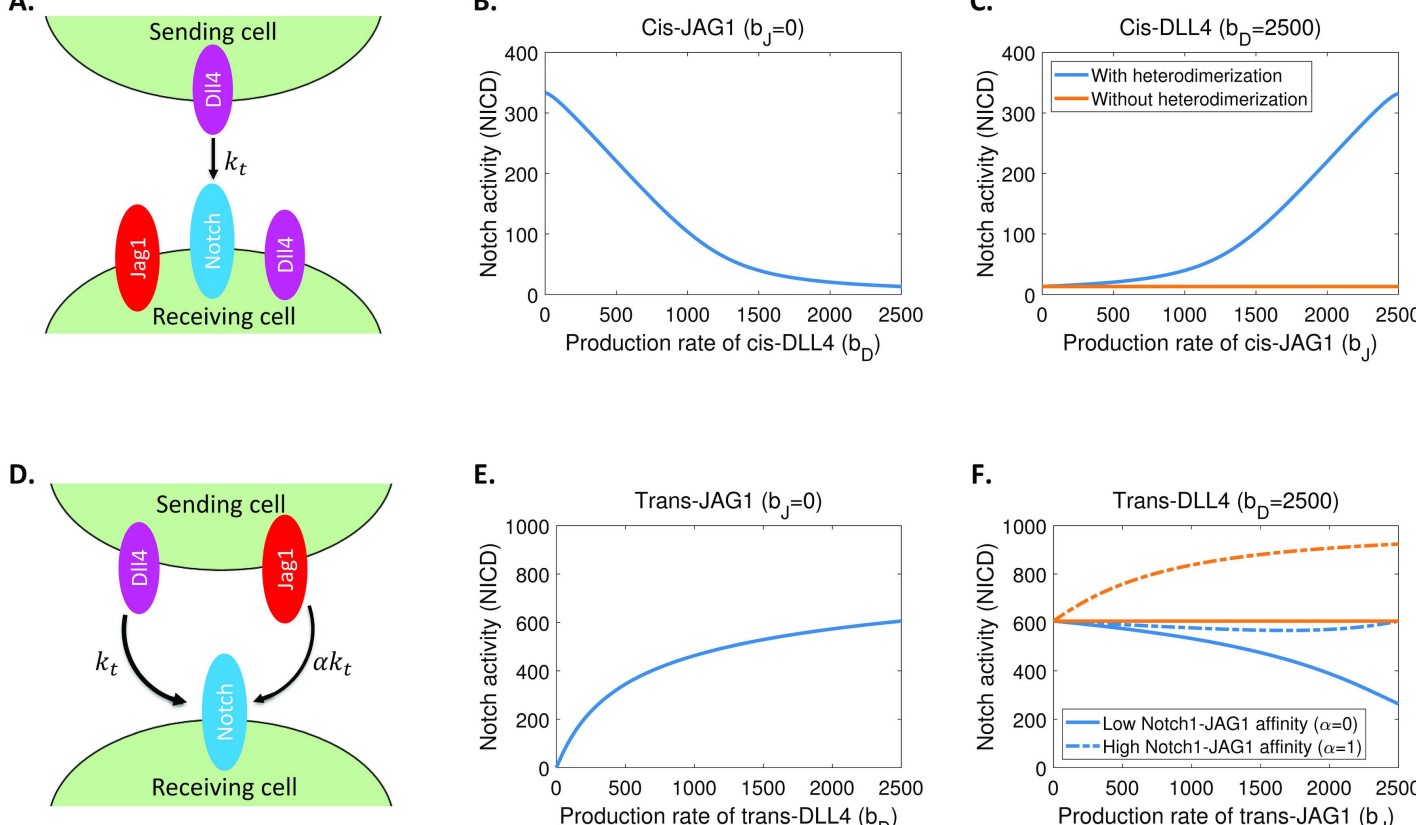

**Fig 4. The potential mechanisms controlling the opposing effects of DLL4 and JAG1 on Notch1 receptor cis-inhibition and trans-activation.** (A) Diagrammatic representation of modelling simulation assays to explore the roles of DLL4 and JAG1 in Notch1 receptor cis-inhibition. (B) Notch activity in the receiving cell as a function of cis-DLL4 production rate. (C) Notch activity in the receiving cell as a function of cis-JAG1 production rate for a high level of cis-DLL4 (the maximum production rate shown in panel B). (D) Diagrammatic representation of modelling simulations assays to explore the roles of DLL4 and JAG1 in Notch1 receptor trans-activation. (E) Notch activity in the receiving cell as a function of trans-DLL4 production rate. (F) Notch activity in the receiving cell as a function of trans-JAG1 production rate for a high level of trans-DLL4 (the maximum production rate shown in panel E). In panel C and panel F, the blue lines represent the scenario with DLL4-JAG1 heterodimerization in the cells, the orange lines represent the scenario without DLL4-JAG1 heterodimerization, the solid lines represent that Notch1 has lower affinity for trans-JAG1 than trans-DLL4, and the dotted lines represent that Notch1 has the same affinity for trans-JAG1 and trans-DLL4.

DLL4-JAG1 heterodimerization, faithfully reproduced our experimental observations (see Figs 2 and 3), which showed that JAG1 blocks DLL4-mediated Notch receptor cis-inhibition. Therefore, the model supports the view that DLL4-JAG1 heterodimerization is necessary to enable the opposing effects of DLL4 and JAG1 on Notch receptor signalling (see S4 and S5A–C Figs for additional simulations).

Having established the requirement of DLL4-JAG1 heterodimerization for DLL4 and JAG1 control of Notch receptor cis-inhibition, we next addressed both the role of DLL4-JAG1 heterodimerization and also the different affinities for Notch1 of trans DLL4 and JAG1 in Notch receptor transactivation. Fig 4D schematically depicts the simulation setup wherein the level of Notch in the receiving cell is fixed and the sending cell could express both DLL4 and JAG1. Fig 4E shows that trans-DLL4, as expected (see Fig 1), efficiently stimulated Notch receptor signalling. Although, *in vivo*, the pattern of ligand expression suggests that leading edge tip cells express only high levels of DLL4 and lack JAG1 expression, neighbouring cells do express both DLL4 and JAG1 ligands [18], and previous reports have investigated co-expression of trans DLL4 and trans JAG1 in the sending cell [18]. In Fig 4F (also S4C–F and S5D–H Figs) we show that model (1), which includes the effects of DLL4-JAG1 heterodimerization (blue lines in Fig 4F), and relatively lower Notch1-JAG1 affinity compared to Notch1-DLL4 affinity (solid lines in Fig 4F), consistent with our experimental data (see Fig 1), faithfully reproduced the reported experimental data (see Fig 6 in [18]), which show that JAG1 limits DLL4-mediated Notch1 receptor trans-activation.

Collectively, our modelling results recapitulate the opposing effects of DLL4 and JAG1 on Notch1 signalling and highlights the importance of a novel mechanism whereby JAG1 directly blocks DLL4-mediated Notch cis-inhibition via DLL4-JAG1 heterodimerization, which could induce a mutual inactivation of DLL4 and JAG1 in cells (see S3 Fig), coupled to the relatively higher affinity for Notch1 of trans DLL4 compared to trans JAG1.

## Recapitulating endothelial cell patterning in sprouting angiogenesis

The preceding analyses considered Notch1 signalling mechanisms controlled by DLL4-JAG1 heterodimer formation at the molecular and individual cell level. We next tested the validity of our model in the context of endothelial cell fate in sprouting angiogenesis. To mimic the *in vivo* microenvironment, we considered a gradient of VEGF across multiple connected cells yielding a front region with high VEGF concentrations (see Modelling methods), which delimited the site of tip/stalk cell formation through activation of the DLL4/JAG1-Notch-VEGF signalling network. In Fig 5A, we performed numerical simulations using different production rates of DLL4/JAG1 in the cells; and graphical depictions of the steady state levels of DLL4, JAG1, Notch activity and VEGFR2 activity in these interacting cells are shown. Our simulations show functional differentiation of endothelial cells into tip cells (characterized by high VEGFR2 activity) and stalk cells (with low VEGFR2 activity) over a broad range of DLL4/JAG1 production rates (see Figs 5A and S6). These results recapitulate the observed *in vivo* endothelial cell patterning in angiogenesis [18,26,31] where the tip cells and stalk cells are arranged alternately at the leading edge of the sprouting vasculature, which have also been successfully reproduced by other mathematical/computational models [26–29]. Furthermore, our model recapitulates the reported spatial expression of DLL4 and JAG1 in the endothelial cells [18], where the tip cells at the growing front have high levels of DLL4 (low levels of JAG1) whilst trailing cells express relatively lower levels of DLL4 (and higher levels of JAG1).

In contrast to the simulations which include DLL4-JAG1 heterodimerization, the expected tip/stalk cell differentiation could not be reproduced in simulations without DLL4-JAG1 heterodimerization (see Fig 5B), although this model could recapitulate some features of the

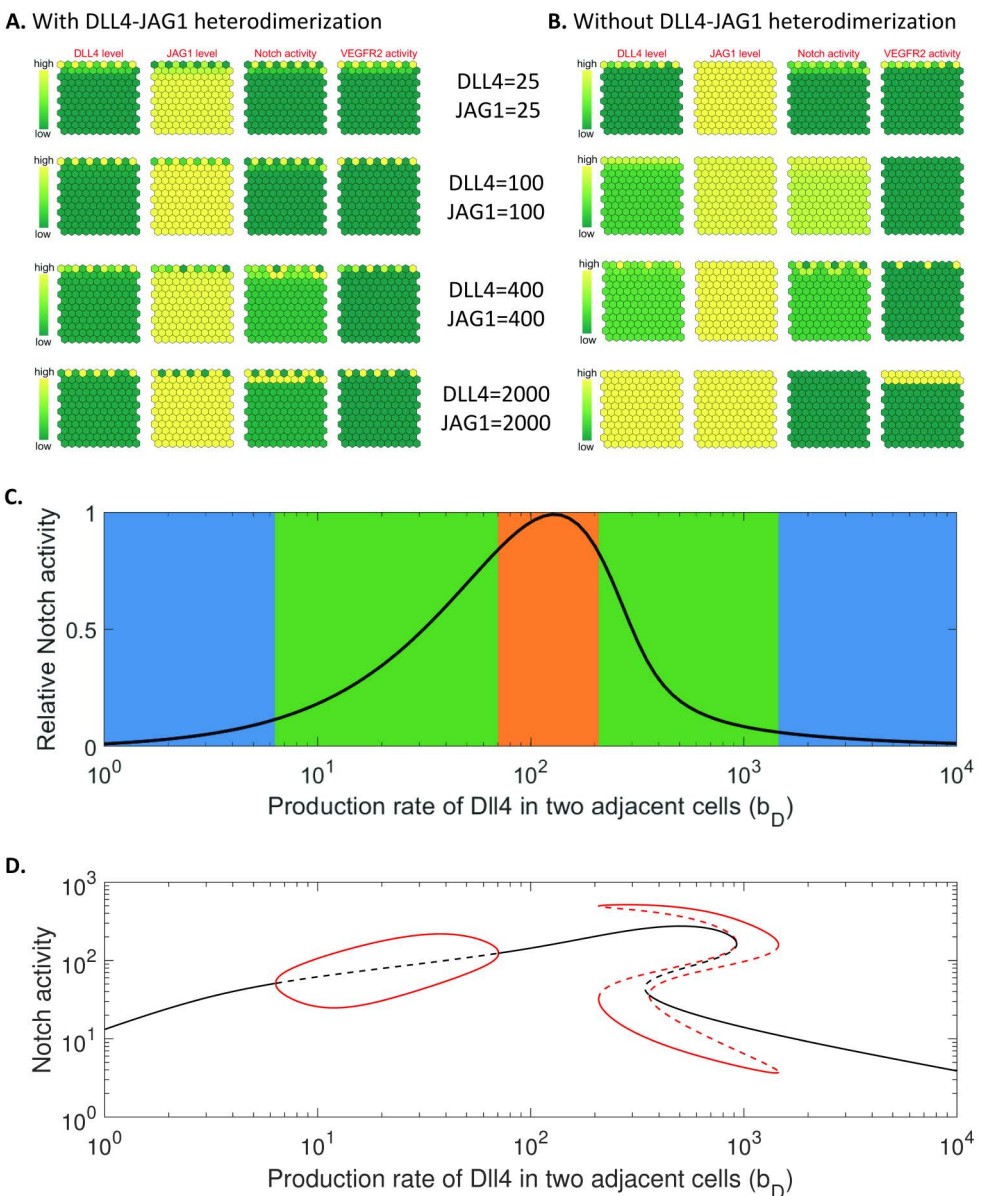

**Fig 5. Recapitulating multicellular patterning (tip/stalk cell formation) at the early stage in sprouting angiogenesis.** (A and B) In silico replication of multicellular patterning mediated by DLL4, JAG1, Notch activity or VEGFR2 activity along the front edge of sprouting angiogenesis for different production rates of DLL4 and JAG1. The levels of VEGF decay exponentially from the top to the bottom (for a movie of the simulations, see S1 and S2 Videos). (C and D) Notch activity in either of two interacting cells as a function of DLL4 production rate in cells. The simulations are based on the DLL4/JAG1-Notch signalling network (model (1), panel C) or DLL4/JAG1-Notch-VEGF signalling network (model (2), panel D) by assuming that there is no DLL4-JAG1 heterodimerization in the cells. The solid (black and red) lines represent stable states of VEGFR2 activity, and the dotted lines represent the unstable states of VEGFR2 activity.

multicellular patterning observed in Fig 5A at a subset of DLL4/JAG1 production rates (e.g., see DLL4 = JAG1 = 25 and DLL4 = JAG1 = 400 in Fig 5B). To gain a deeper understanding of the necessary conditions for the differentiation of neighbouring cells into more specialized tip/stalk cells, from a modelling perspective, we present the steady state Notch activity as a

function of DLL4 production rate by simulating two interacting cells in the absence of DLL4-JAG1 heterodimerization (Fig 5C, model (1); Fig 5D, model (2)). Comparing Fig 5C and 5D, if the intracellular Notch activity was within a defined range (marked by green shading in Fig 5C), the DLL4-Notch-VEGF signalling network would induce multiple stable states of proteins in the cells (the solid red lines in Fig 5D), which corresponds to the emergence of the different cell fates of the interacting cells (see DLL4 = JAG1 = 25 and DLL4 = JAG1 = 400 in Fig 5B). In Figs 2–4, we have shown that the heterodimerization of DLL4 and JAG1 could be a central mechanism in the control of Notch1 signalling. Thus, these simulations suggest that the DLL4-JAG1 heterodimerization could be critical for tip/stalk cell formation during sprouting angiogenesis within specified thresholds of Notch ligand expression.

## Modelling the molecular basis of the opposing effects of DLL4 and JAG1 on endothelial cell fate in sprouting angiogenesis

In the simulations of tip/stalk cell patterning in sprouting angiogenesis, our DLL4/JAG1-Notch-VEGF signalling model (2) recapitulates the reported finding [18] that a tip cell (characterized by high VEGFR2 activity) expresses high levels of DLL4 and lacks JAG1 at the growing front (see Fig 5A). We next explored the mechanisms by which differential DLL4 and JAG1 production rates could underlie their opposing effects on sprouting angiogenesis (see S7 Fig A for a schematic depiction of this process). In Fig 6A and 6B, we simulated a line of interacting cells, representing the growing front of sprouting angiogenesis, for varying production rates of DLL4 and JAG1 in the cells, respectively. Because tip cells direct the emergence of new sprouts during sprouting angiogenesis [1], the effects of Notch ligands (DLL4 and JAG1) on sprouting angiogenesis could be characterized by their effects on tip cell number. According to experimental estimates (see Parameters in the Methods section), when the production rates of DLL4 and JAG1 correspond to the wild-type scenario, in response to increased VEGFR activity, tip cells express high levels of DLL4 and lack JAG1 (WT in Fig 6A and 6B). For low production rates of DLL4 ( $DLL4^{+/-}$ in Fig 6A) or high production rates of JAG1 ( $JAG1^{iGOF}$ in Fig 6B), the modelling predicts that endothelial cells would preferentially adopt a tip cell fate (characterized by high VEGFR2 activity) and thus could be expected to form an excessive number of sprouts. For high production rates of DLL4 ( in Fig 6A) or low production rates of JAG1 ( $JAG1^{+/-}$ in Fig 6B), the modelling predicts that the endothelial cells would preferentially adopt a stalk cell fate (characterized by low VEGFR2 activity) and thus could be expected to develop a sparse vascular network. These simulations clearly demonstrate the opposing effects of DLL4 and JAG1 on endothelial cell fate, and the biological implications are consistent with experimental observations in sprouting angiogenesis [18,24,30–32]. To decipher the molecular basis of the opposing effects of DLL4 and JAG1 on endothelial cell fate, we presented the steady-state levels of VEGFR2 activity in two interacting cells as a function of DLL4 production rates for different production rates of JAG1 in the presence (Fig 6C) or absence (S7B Fig) of DLL4-JAG1 heterodimerization. The numerical analysis in Fig 6C and S7B Fig suggests that the opposing effects of DLL4 and JAG1 on endothelial cell fate is highly dependent on DLL4-JAG1 heterodimerization because the bifurcation diagram of VEGFR2 activity would be independent of JAG1 production rate in the cells if there was no DLL4-JAG1 heterodimerization (see S7B Fig).

Recently, Tiemeijer et al. [33] utilized an elegant combination of bioengineering and a novel endothelial cell culture assay to study endothelial sprouting controlled by defined patterns of JAG1 and DLL4. They showed that defined stripes of functional DLL4 protein guide the location and direction of angiogenic sprouts by specifically excluding spouting within the stripes of DLL4, while equivalent stripes of functional JAG1 protein do not exert such effects, which is depicted schematically in Fig 7A. Fig 7B and 7C shows that our DLL4/

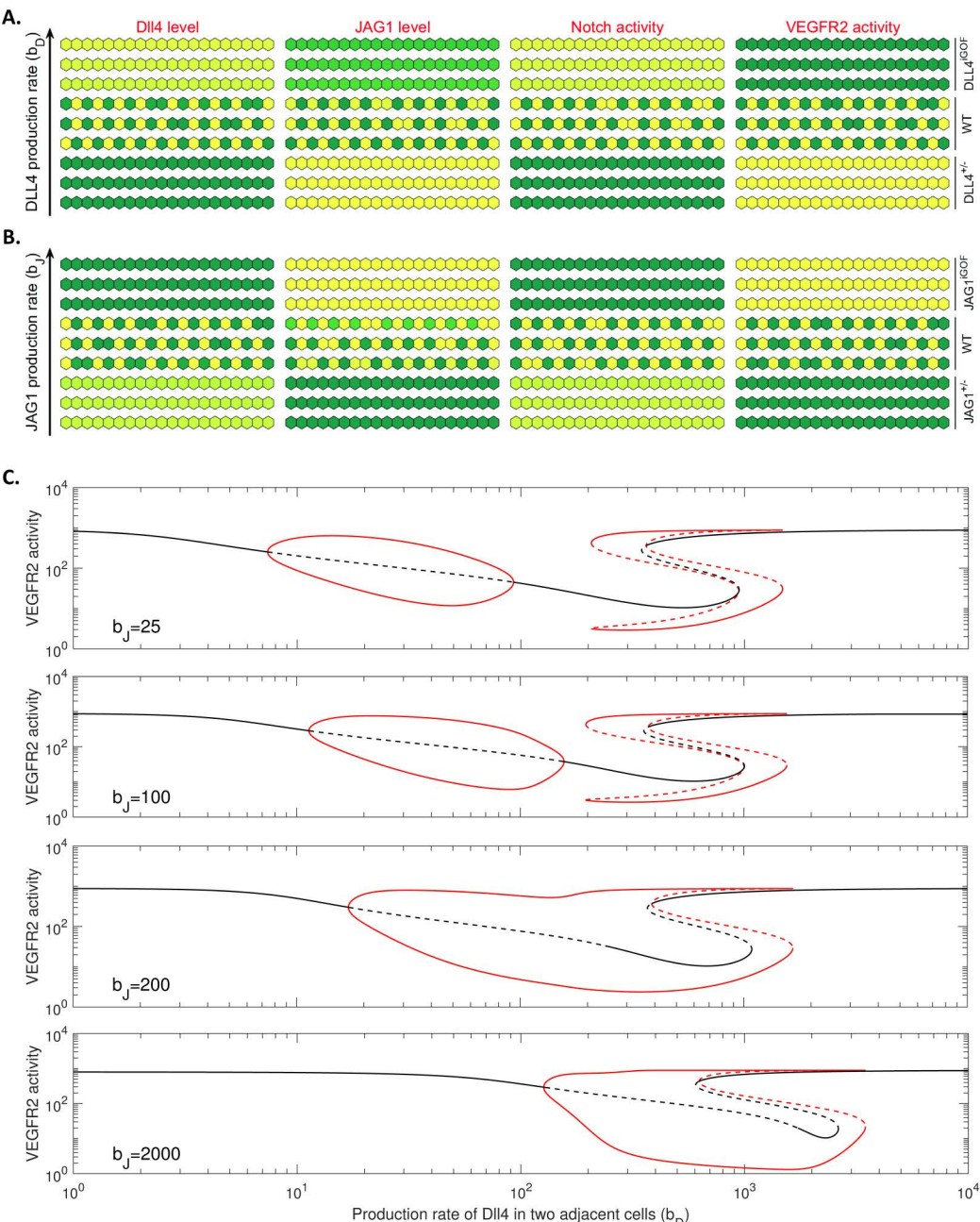

**Fig 6. Recapitulating the opposing effects of DLL4 and JAG1 on endothelial cell fate in sprouting angiogenesis.** (A and B) Numerical simulations of DLL4 levels, JAG1 levels, Notch activity and VEGFR2 activity in a line of interacting cells. The production rates of DLL4 and JAG1 increase from the bottom to the top. A warmer color means higher levels of corresponding proteins in the cells. According to the production rates of DLL4 and JAG1, we divided the simulations in panels A and B into mutant groups ($DLL4^{+/-}$ or $JAG1^{+/-}$), wild-type groups (WT), and overexpression groups ($DLL4^{iGOF}$ or $JAG1^{iGOF}$). (C) Bifurcation diagram of VEGFR2 activity in either of two interacting cells driven by DLL4 production rate in cells for different JAG1 production rate. The simulations are based on the DLL4/JAG1-Notch-VEGF signalling model (2) with DLL4-JAG1 heterodimerization.

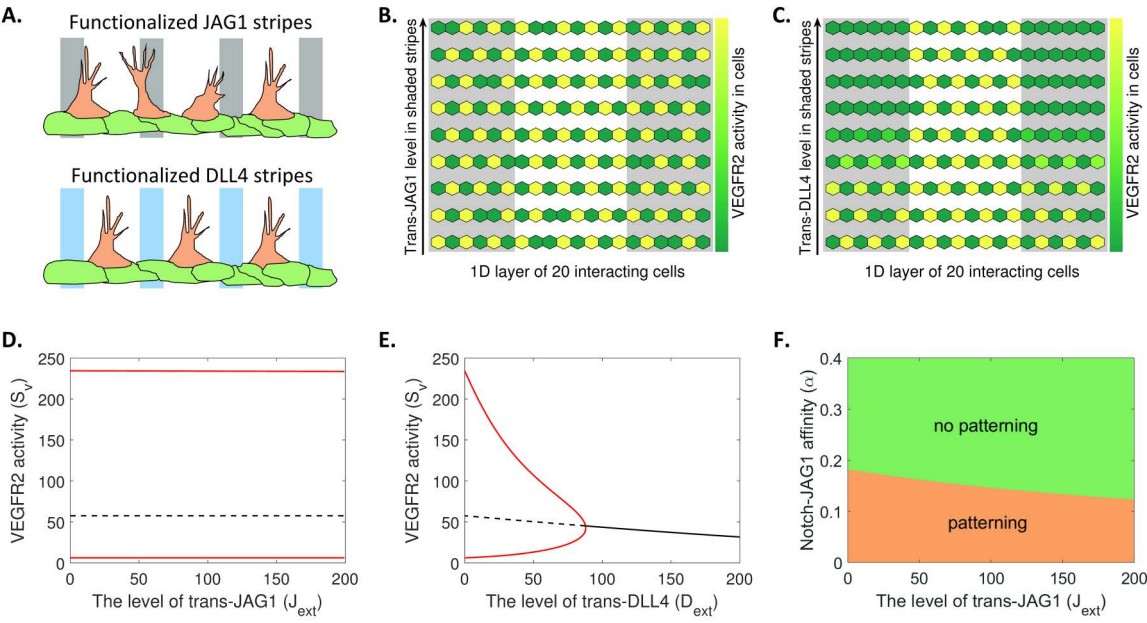

**Fig 7. Recapitulating the spatial control of DLL4 and JAG1 to sprouting angiogenesis.** (A) Diagrammatic representation of the effect of trans-JAG1 and trans-DLL4 on endothelial cell behaviours. DLL4 elicits spatial control in sprouting angiogenesis whereas JAG1 has no effect on it. (B and C) Numerical simulations of VEGFR2 activity in a line of interacting cells exposed to different levels of trans-JAG1 (B) or trans-DLL4 (C) engineered in two (shaded) stripes. The level of trans-ligands in the (shaded) stripes increases linearly from the bottom to the top. (D and E) Bifurcation diagram of VEGFR2 activity in either of two interacting cells driven by the level of trans-JAG1 (panel D) or trans-DLL4 (panel E) exposed to the cells. Here the affinity of Notch1 for trans-JAG1 is low ($\alpha = 0$). (F) Phase-diagram (two parameters bifurcation) of cell fate of interacting cells driven by the trans-JAG1 level and JAG1-Notch affinity. Patterning means that the interacting cells would show alternating cell fate marked by the proteins in the cells (as shown in panels B and C).

JAG1-Notch-VEGF signalling model (2) could recapitulate these effects. We simulated a line of interacting cells exposed to increasing concentrations of either trans-JAG1 (Fig 7B) or trans-DLL4 (Fig 7C). In agreement with the findings of Tiemeijer et al. [33], increasing levels of trans-DLL4, but not trans-JAG1, spatially restricted endothelial cell patterning. In Fig 7D and 7E, we present the steady-state levels of VEGFR2 activity in two interacting cells driven by different levels of trans-JAG1 and trans-DLL4. Clearly, high levels of trans-DLL4 lead to reduced VEGFR2 activity in the cells (Fig 7E), while changes in trans-JAG1 levels had little effect on VEGFR2 activity (Fig 7D). Consistent with our experimental results (Fig 1) and previous reports [33], the different effects of DLL4 and JAG1 in controlling sprouting angiogenesis is also highly dependent on the different affinities for Notch1 of trans DLL4 and JAG1 (see Fig 7F).

Collectively, these data demonstrate that our model recapitulates endothelial cell fate driven by the Notch ligands, DLL4 and JAG1, in sprouting angiogenesis, and underscore the importance of DLL4-JAG1 heterodimerization and the different affinities for Notch1 of trans DLL4 and JAG1 in this process.

## Discussion

Since the discovery of the Notch receptor in the second decade of the last century, both pioneering work in *in vivo* model systems, as well as cell and biochemical analyses, have provided a detailed, albeit incomplete, picture of its essential roles in animal development and the basic molecular underpinnings of its function. In general outline, Notch receptors

can interact with their ligands within the same cell or between cells and the net effect of these interactions determines the level of Notch receptor activation. Related to this, a currently unanswered question of central importance is how different Notch receptor and ligand combinations control this process to produce its observed effects. Given the wide range of distinct physiological phenomena which are governed by this relatively simple receptor-ligand network (see [1–7]), multiple additional regulatory mechanisms must operate to enable the broad repertoire of possible responses, including biomechanical forces [4,8], post-translational modifications of both receptors and ligands [7], asymmetric cell divisions and processes which promote and control action-at-a-distance Notch signalling activities [34].

In this study, we have used simple experimental approaches coupled to mathematical modelling to address this question, specifically by considering how the JAG1 and DLL4 ligands differentially control Notch1 receptor activation, a question of fundamental relevance in light of the well-established evidence for the opposing effects of DLL4 and JAG1, which together control angiogenic sprouting [18,19]. Consistent with previous studies [35–37], our data show that, by comparison with JAG1, DLL4 has a significantly greater capacity to stimulate trans-activation of Notch1 receptor signalling (see Fig 1). We further show that DLL4, but not JAG1, efficiently mediates Notch1 receptor cis inhibition. In addition to the significant differences in the magnitude of Notch1 transactivation and cis inhibition elicited by each ligand (Fig 2), we have uncovered the previously overlooked phenomenon of ligand heterodimerization (Fig 3). Specifically, we show that a JAG1/DLL4 heteromeric complex blocks DLL4-dependent Notch receptor cis inhibition thereby allowing receptor transactivation, which is stimulated by ligands expressed by neighbouring cells. In this view, JAG1 can directly, and precisely, fine-tune DLL4 transactivation of Notch1, which is determined by the relative expression levels of each ligand. Given the evidence that DLL4 homodimerization could be necessary for cis inhibition [15], one plausible mechanism to account for our findings is that JAG1 monomers compete with DLL4 for binding to other DLL4 monomers. In this respect, it is noteworthy that JAG1 appears to display a higher affinity for DLL4 than DLL4 has for itself (see S8 Fig). Our data further suggest that the absence of a VWFC domain in JAG1, an oligomerization domain which is expressed by the highly related JAG2 ligand, has enabled this mechanism by precluding the formation of JAG1 homodimers (in contrast to all other Notch ligands (see Fig 2; [15]). It is compelling that there are parallels between this (presumably relatively ancient) mechanism and a remarkable recent evolutionary innovation which has been proposed to explain, at molecular resolution, the dramatic expansion in size of the cortical region of the human brain, whereby small ligand-like Notch2 receptor paralogues enable Notch receptor activation by directly binding to, and potentially blocking, DLL1-dependent receptor cis inhibition [38].

Significant advances in our understanding of Notch signalling have emerged, and continue to emerge, through use of powerful mathematical approaches coupled with *in silico* modelling of the biological consequences of the molecular and cellular processes involved in this pathway [14–16,25–29,39–43]. Several existing models have addressed the distinct roles of JAG1 and DLL4 in angiogenic patterning [33,41]. Recently, using a combined experimental and computational approach, Tiemeijer et al. [33] showed that spatial and temporal differences in the control of Notch activity by DLL4 and JAG1 underlie endothelial sprouting. Boareto et al. [41] undertook a different approach to model the asymmetric effects of DLL4 and JAG1 on endothelial cell fate in which the opposing effects of DLL4 expression and JAG1 expression on Notch signalling suggest that high production levels of DLL4 establish the tip-stalk phenotype in interacting cells while high production rates of JAG1 destabilizes the tip and stalk cell fates and can give rise to a hybrid tip/stalk phenotype.

Here, we have proposed a new mathematical model, which is a refinement of our previously described general model [15], which takes into account our new experimental insights into DLL4 and JAG1 control of Notch1 receptor signalling. It should be stated that we cannot rule out that ligand monomers and dimers might dynamically associate and dissociate during Notch signalling, but most mathematical models [14,15,17,28,29], including the model we proposed here, do not take into account the possible reversibility of these interactions. However, we can show that if ligand dimerization, as well as ligand/receptor cis- and trans-binding, is reversible, the conclusions of this paper will still hold (see Modelling methods). Our model faithfully recapitulates both published data (see Fig 4D–F) and the experimental findings of the study presented here (see Fig 4A–C). Numerical simulations and bifurcation analysis of the model predict that the high affinity of DLL4, and low affinity of JAG1, for Notch1 is critical for recapitulating the distinct responses of endothelial cells to trans DLL4 and JAG1 (Fig 7A–F), in agreement with other models [16]. Importantly, our model underscores the role of DLL4-JAG1 heterodimerization in determining the opposing effects of DLL4 and JAG1 on angiogenesis, specifically, the specialization of endothelial cells into leading edge tip cells and trailing stalk cells, which is necessary for vessel sprouting (Figs 5 and 6).

In this model, we have focussed on endothelial cells and the two principal signalling pathways, Notch1 and VEGFR signalling, which mutually guide endothelial cell fate. A future elaboration of this model could incorporate other Notch receptor members, and other cytokine/growth factor signal transduction pathways (e.g., the inflammatory cytokine, tumour necrosis factor-$\alpha$ (TNF-$\alpha$) was reported to up-regulate JAG1, but reduce DLL4 transcript levels in endothelial cells [18]. It could also take account of other cell types and signalling modules, including the Notch3 signal transduction pathway in vascular smooth cells [19]. Related to that, whilst negative feedback mechanisms have been shown to be critical for angiogenesis, specifically by linking VEGFR signalling to Notch via VEGF stimulation of DLL4 expression, and Notch-mediated suppression of VEGFR expression, Page et al. [44] delineated a positive feedback mechanism, which could underpin the timing and the scale of angiogenesis. Moreover, during Drosophila development, there is evidence that Delta ligand in the sending cell activates Notch in the receiving cell which suppresses Delta expression in the receiving cell (see [34]). In the context of our new model, by considering these facets of Notch signalling and endothelial sprouting, we could get not only deeper insights into normal developmental sprouting angiogenesis but also into other physiological processes, such as wound healing [19], and the pathological angiogenesis unique to solid tumours, and chronic inflammatory and autoimmune diseases [45,46]. Indeed, regarding other Notch receptor family members, our evidence that ligand heterodimerization could play a role in Notch signalling has potentially important implications for our understanding of the control of this central signal transduction pathway more generally. For example, heteromeric Notch ligand complex formation might serve to increase the number of possible Notch receptor signalling outputs. Certainly, our reported effects are not limited to Notch1 signalling alone. In comparable assays of receptor transactivation and cis inhibition, JAG1/DLL4 heterodimers similarly controlled both Notch2 and Notch3 signalling activity (see S9 Fig). Since all Notch ligands share a common overall architecture, it could be of interest to investigate if other Notch ligands can form heteromeric complexes, and to decipher the possible biological consequences of these interactions. Additional mechanisms could also contribute to the overall regulation of the Notch network including signalling initiated by processing of the ligand intracellular domains [47], and a possible role for Notch receptor oligomerization/dimerization, which would add an additional layer of complexity to the system, though the precise molecular details of this remain elusive [48].

As well as angiogenesis, our results may shed light on our understanding of the patterning of other tissues. A well-characterized, ancient example is dorsal/ventral (D/V) boundary formation in Drosophila wing disc development. Dorsal cells express Notch, Delta and Serrate, ventral cells express Notch and Delta only, moreover, Delta expression is especially high at the D/V boundary [49–52], an arrangement not dissimilar to the expression patterns of DLL4 and JAG1 found in tip and stalk cells in the angiogenic front of the developing mouse retina [18]. At the early stage of dorsal/ventral boundary formation, Serrate in dorsal cells is believed to transactivate Notch in ventral cells and Delta in ventral cells transactivates Notch in dorsal cells, thereby delimiting a sharp stripe of Notch activation at the dorsal/ventral interface. Currently, it is not fully understood how such a defined pattern of Notch activity is generated and sustained, and a possible contributing factor could be that the relative production rates of Serrate and Delta in the two cell types establishes signalling thresholds, comparable to those observed in adjacent endothelial tip and stalk cells, which enables a heteromeric ligand complex to spatially and temporally restrict Notch receptor activation.

In summary, we have presented a model of Notch signalling which incorporates the novel finding that JAG1 directly antagonizes the function of DLL4 through the formation of heteromeric DLL4-JAG1 complexes, which could underlie the observed opposing effects of JAG1 and DLL4 in sprouting angiogenesis. By integrating DLL4 and JAG1 dynamics, and DLL4/JAG1 dimerization, into our previous theoretical understanding of Notch signalling, a new mathematical model for tip endothelial cell selection in angiogenesis is proposed which successfully recapitulates the biological observations that JAG1 and DLL4 play distinct roles in angiogenesis. Bifurcation analysis of the mathematical model predicts that DLL4-JAG1 dimerization and their respective asymmetric affinities for Notch are critical for explaining the opposing effects of DLL4 and JAG1 on angiogenesis. Future work will determine if our findings may have broader relevance to other normal and pathological tissue processes.

## Materials and methods

### Modelling methods

In this paper, we present a mathematical model (1) of Notch1 signalling, which does not consider the possible reversibility of ligand and receptor interactions. In contrast to previous models [14,15], DLL4-JAG1 heterodimerization is a newly introduced protein-protein interaction. Reversible DLL4-JAG1 heterodimerization is represented mathematically by:

$$D + J \underset{\text{association/dissociation}}{\longleftrightarrow} \left[DJ\right] \xrightarrow{\text{degradation}} \varnothing.$$

If the DLL4-JAG1 complexes $\left[DJ\right]$ can dissociate, the dynamics of DLL4 levels and JAG1 levels can be expressed as:

$$\frac{dD}{dt} = b_D - \beta D - 2k_{ho}D^2 - \left(k_+ DJ - k_- \left[DJ\right]\right) - k_t N_{ext} D,$$

$$\frac{dJ}{dt} = b_J - \beta J - \left(k_+ DJ - k_- \left[DJ\right]\right) - \alpha k_t N_{ext} J, \tag{9}$$

$$\frac{d\left[DJ\right]}{dt} = \left(k_+ DJ - k_- \left[DJ\right]\right) - \beta \left[DJ\right],$$

where $k_+$ and $k_-$ represent the association/dissociation rates of DLL4-JAG1 complexes $\left[DJ\right]$. At the positive steady state ($\bar{D}$, $\bar{J}$, $\overline{\left[DJ\right]}$) of Eq. (9), we have:

$$b_D - \beta\bar{D} - 2k_{ho}\bar{D}^2 - \left(k_+\overline{DJ} - k_-\overline{[DJ]}\right) - k_t N_{ext}\bar{D} = 0,$$
$$b_J - \beta\bar{J} - \left(k_+\overline{DJ} - k_-\overline{[DJ]}\right) - \alpha k_t N_{ext}\bar{J} = 0, \qquad (10)$$
$$k_+\overline{DJ} - k_-\overline{[DJ]} - \beta\overline{[DJ]} = 0.$$

Let $k_{he} = \beta k_+ / (\beta + k_-)$, the first two equations in Eq. (10) can be rewritten as:

$$b_D - \beta\bar{D} - 2k_{ho}\bar{D}^2 - k_{he}\overline{DJ} - k_t N_{ext}\bar{D} = 0,$$
$$b_J - \beta\bar{J} - k_{he}\overline{DJ} - \alpha k_t N_{ext}\bar{J} = 0, \qquad (11)$$

Comparing Eq. (11) with the first two equations of model (1), it can be seen that introducing DLL4-JAG1 heterodimerization reversibility does not change the results of this paper. More generally, if the cis- and trans- binding of Notch ligands and receptors is also assumed to be reversible, the equilibrium of an extended model does not differ substantially from the current, widely accepted, simplified model [14]. In S3 Fig, we also show that allowing the disassociation of DLL4 homodimers does not change the predicted mutual inactivation between DLL4 and JAG1 in the same cell.

**Parameters.** The values of parameters in our model are determined based on previous works and our new data. Published data have suggested that Notch-related protein levels vary by up to a few hundred ng/ml, or a few thousand molecules per cell [53], thus the production rates of Notch ligands, Notch receptor, VEGFR2 are assumed to produce corresponding amounts of protein in the cell (see Table 1). The degradation rate of proteins is set $\beta = 0.1$ /hour, equivalent to a half-life of about 7 hours [54]. The degradation of free Notch Intracellular Domain ($S$) and free VEGFR2 Intracellular Domain is assumed to be $\beta_S = 0.5$ /hour because the signal in Notch signalling and VEGFR2 signalling decays rapidly [55]. In S5 Fig, we tested different degradation rates of proteins, and the variation in degradation rates did not affect our results. We assume that the DLL4-JAG1 heterodimerization rate is $k_{he} = 10^{-3} / (\text{molec} \star \text{hour})$, which is bigger than the DLL4-DLL4 homodimerization rate $k_{ho} = 10^{-4} / (\text{molec} \star \text{hour})$ [15] because our new data imply that DLL4-JAG1 affinity could be bigger than DLL4-DLL4 affinity (see S8 Fig). The trans-activation rate, cis-inhibition rate and VEGF-VEGFR2 binding rate were derived from previous studies [15,28,29], the authors estimated their parameters by referring to relevant experimental reports. The fold change of VEGFR2 activity-dependent up-regulation for DLL4 expression is close to 10 [23], and the fold change of Notch activity-dependent regulation for VEGFR2 expression is assumed to be 0 [28,41]. The values of other parameters related to VEGFR2/Notch activity-dependent regulation for DLL4 and VEGFR2 also come from previous modelling studies [28,29,41].

**Numerical simulations.** Unless otherwise specified, the numerical simulations were performed on MATLAB R2021a using the parameter values in Table 1. The fourth-order Runge–Kutta methods were used in all scenarios.

In the simulations, the levels of trans-ligands $D_{ext}$, $J_{ext}$ are context-dependent. For the single cell in S3 Fig, $D_{ext} = J_{ext} = 0$; for multiple interacting cells (including two cells), the level of trans-ligand is the average of ligands in neighbouring cells (Figs 4, 5, 6 and S5); for multiple interacting cells exposed to fixed level of trans-ligands, the level of trans-ligand is the average level of ligands in neighbouring cells plus the level of external ligands which varies from 0 to 200 (Fig 7).

In Figs 5A, 5B and S6, each cell is represented as a regular hexagon with side length 1. The concentrations of VEGF decay exponentially from the growing front $y_0$ to trailing areas, according to $V(y) = V_{ext} / \exp(a|y - y_0|)$, where $y$ is the vertical coordinate of each cell center.

In particular, $y = y_0$ represents the vertical coordinate of the center of the first row of cells. Parameter $a$ controls the sharpness of VEGF concentrations ($a = 1$ in Fig 5). The initial value of these simulations is random, and the boundary value uses the Dirichlet boundary 0.

In Figs 6 and 7, the line of cells represents the growing front, thus the concentration of VEGF is fixed at $V_{ext}$.

**Numerical bifurcation analysis.** In Figs 5D, 6C and 7D–F, the numerical bifurcation analysis for two cell system was performed by using the package MatCont [56]. The black solid lines represent the stable homogeneous steady-states, the black dotted lines represent the unstable homogeneous steady-states, the red solid lines represent the stable heterogeneous steady-states, and the red dotted lines represent the unstable heterogeneous steady-states.

## Experimental methods

**Cell culture, biochemistry and molecular biology.** Human embryonic kidney 293T cells were cultured in DMEM (Gibco) supplemented with 10% fetal bovine serum (Gibco). Cell lines were typed using short tandem repeat analysis of the DNA and all cell lines were checked for mycoplasma with the MycoAlert kit (Lonza). Transfections, lentivirus production and cell infections, Western blotting and co-immunoprecipitations have been described previously [57,58]. All lysis buffers contained a cocktail of protease inhibitors (phenylmethylsulfonyl fluoride, trypsin inhibitor, pepstatin A, leupeptin, aprotinin).

**Recombinant protein production/ *in vitro* protein:protein interaction.** Domains for recombinant protein production were cloned into the pET 28a vector in-frame to an N-terminal 6x HIS epitope. His epitope–tagged proteins were manufactured in *Escherichia coli* BL21(DE3). Following sonication (Misonix Sonicator 3000) in 3 mls ice-cold buffer/ 50 ml bacterial culture (150 mM NaCl, 2.7 mM KCl, $Na_2HPO_4$, $KH_2PO_4$, 20 mM imidazole, 10 mM $\beta$-mercaptoethanol), proteins were purified onto 50 ul of Nickel- agarose beads (Qiagen) by 3 hours rolling at 4C. Beads were washed in 10 x 1 ml of the same buffer. Protein yields were determined by Bradford assay (Bio-Rad) and relative protein integrity and purity was determined by SDS-PAGE and Colloidal Blue staining (Invitrogen). Purified recombinant protein was incubated with 10 ul nickel beads in 1 ml of buffer for 2 hours at 4°C with *in vitro* translated DLL4 proteins made using the TNT-coupled reticulocyte *in vitro* translation system (Promega). Beads were washed x10 with 1 ml of buffer. Proteins were separated by SDS-PAGE and associated proteins were detected by Western blot.

**Plasmid construction.** Unless otherwise stated, all cDNAs were fused in-frame with a C-terminal FLAG, HIS or HA epitope tag and were cloned into the pLV lentiviral vector and pCS2 expression plasmid. Expression of these proteins was determined using antibodies directed against either the coding region of the protein or the epitope tag. Mutants were generated by site-directed mutagenesis using Phusion High-Fidelity DNA polymerase (Thermo fisher). All constructs were verified by Sanger sequencing (Macrogen).

**Luciferase reporter.** Stable cell lines expressing epitope-tagged ligands and receptors were established via infection of cells with lentiviruses harbouring the appropriate ligand/receptor cDNA followed by selection with the encoded antibiotic (puromycin or neomycin). For 'cis inhibition' assays, cells co-expressing ligand, receptor and reporter were co-cultured with cells expressing ligand alone (to enable transactivation). Comparable results were obtained for co-culture ratios of 1:1, 1:2, 1:4. For transactivation analyses, cells stably expressing ligand alone were co-cultured with cells expressing receptor and reporter (1:1 ratio) at a confluency of approximately 90% (0.45 x $10^6$ cells/well), to ensure cells were touching to enable suitable transactivation. For each experiment, cells were seeded in triplicate in 12-well plates. The Notch luciferase reporter harboured 10x RBPJ consensus binding sites, and was co-transfected

with Renilla luciferase control plasmid. Transfection efficiencies (routinely >90%) were determined through visualization of co-transfected plasmid encoding the Tomato fluorescence reporter. Cells were lysed 36 hours post-plating, and luciferase activity was measured using a luciferase assay substrate (Promega). Luciferase activity was normalized by measuring Renilla luciferase activity (Promega). Receptor and ligand protein levels were determined by Western blotting (see S10 Fig). Experiments were performed three times. The statistical hypothesis tests in this study were performed using Student's t-test. We defined $P<0.0001$ as a highly significant difference (**), $P<0.001$ as a significant difference (*), and $P\geq0.001$ as not significant (n.s.).

**Ligand/receptor binding assay.** Confluent 10 cm dishes of 293T cells stably expressing HA-epitope tagged Notch receptors were washed with ice-cold PBS/BSA (1 mg/ml). Cells were stimulated with 50 μL of ligand (0.5 mg/ml) in the presence of 1.5 mls of PBS/BSA (1 mg/ml) for 1 hour, with gently shaking, at 4ºC. Cells were washed once with PBS. Cells were incubated with crosslinkers in the presence of 2 mls of ice-cold PBS for 15 minutes, with gently shaking at 4ºC: 5 μL/ml DSS (54 mML) and 5 μL/ml BS3 (13 mM). Cells were washed once with ice-cold detachment buffer (10mM Tris-HCl pH 7.4, 1mM EDTA pH 7.4, 10% glycerol), scraped off the plates in 1 ml of detachment buffer and transferred to Eppendorf tubes. Cell pellets were lysed in 1 ml of solubilization buffer (125mM NaCl, 10mM Tris-HCl pH 7.4, 1mM EDTA pH 7.4, 1% Triton X-100) on ice for 30 minutes and centrifuged at 13k rpm for 10 minutes. Supernatants were incubated with HA antibody (Covance catalog# MMS-101P, 16B12) overnight at 4ºC. 50 μL of washed (4x) protein A/G beads (50% solution) were added and samples were incubated for 45 minutes at 4ºC. The beads were washed four times with solubilization buffer. 50 μL of sample buffer was added and samples were boiled for 5–10 mins prior to gel electrophoresis.

**Antibodies, proteins and drugs.** Antibodies were obtained from the following sources: FLAG mouse M2 monoclonal (Sigma); anti-HA.11 mouse monoclonal (Covance); anti-HA rabbit polyclonal (Abcam); anti-FLAG rabbit (Sigma); anti-γ-tubulin (Sigma); anti-His (Sigma). Nickel-NTA agarose was purchased from Qiagen. Recombinant human ligands were obtained from: JAG1 protein (Acrobiosystems, Cat. No: JA1-H52H9); DLL4 protein (Acrobiosystems, Cat. No: DL4-H5227); DLL4 protein (AbCam, Cat. No: ab219667).

## Supporting information

**S1 Fig. JAG1 efficiently binds to Notch1 in cis.** 293T cells were co-transfected with the indicated combinations of HA epitope-tagged JAG1 and $HIS_6$ epitope-tagged Notch. Ligands were resolved using an anti-HA antibody and ligand-receptor complexes were determined by Western blotting with the indicated antibodies.
(EPS)

**S2 Fig. Cis DLL4-JAG1 heterodimerization enables efficient trans DLL4-dependent cleavage of Notch.** 293T cells expressing, in cis, the indicated combinations of HA epitope-tagged Notch, FLAG epitope-tagged DLL4 or $HIS_6$ epitope-tagged JAG1, were cultured in the presence or absence of recombinant (trans) DLL4 for the indicated time-course. Cells were cultured in 6 cm dishes and lysed directly in denaturing Laemmli buffer to enable detection of processed Notch. Notch receptor cleavage was monitored by Western blotting with an anti-HA antibody. Expression levels of DLL4, JAG and the housekeeping gene, Vinculin, were determined by Western blotting with the indicated antibodies. DLL4-(EGF7) lacks the EGF-like repeat 7.
(EPS)

**S3 Fig. Mutual inactivation of DLL4 and JAG1 in the same cell.** (A) The simulation is based on the DLL4/JAG1-Notch signalling model (1) with strong DLL4-JAG1 heterodimerization

( $k_{he} = 0.1$ ). The production rate of DLL4 in the cell is fixed at $b_D = 100$ . (B) The simulation was performed under the same conditions, but the dissociation of the DLL4 homodimer was taken into account.
(TIFF)

**S4 Fig. Two-dimensional simulations of the results presented in** Fig 4. The relative levels of Notch activity (NICD) in response to different production rates of: (A and B) cis-DLL4 ( $b_D$ )/ JAG1 ( $b_J$ ), and (C–F) trans-DLL4 ( $b_D$ )/JAG1 ( $b_J$ ), across the two-dimensional plane ( $b_D$ , $b_J$ ).
(TIFF)

**S5 Fig. Extended simulations of the results presented in** Fig 4. (A–C) Notch activity in the receiving cell as a function of cis-ligand production rates. (D–H) Notch activity in the receiving cell as a function of trans-ligand production rates. The degradation rates of ligand monomer, ligand dimer and Notch receptor are independently assumed to be: 0.05/h (a half-life of about 14 hours), 0.15/h (a half-life of about 5 hours), or 0.25/h (a half-life of about 3 hours).
(TIFF)

**S6 Fig. Extended simulations of the cellular patterning described in** Fig 5A. The simulations were performed with a shallower VEGF gradient. The VEGF concentration was set according to: (A) $V(y) = V_{ext} / \exp(2|y - y_0|)$; (B) $V(y) = V_{ext} / \exp(5|y - y_0|)$ ; C) $V(y) = V_{ext} / \exp(35|y - y_0|)$ . See Numerical simulations (Methods) for simulation information.
(TIF)

**S7 Fig. (A) Diagrammatic representation of the effects of JAG1 and DLL4 on endothelial cell behaviours.** DLL4 inhibits sprouting angiogenesis whereas JAG1 promotes it. (B) Bifurcation diagram of VEGFR2 activity in either of two interacting cells driven by DLL4 production rate in the cells. The simulations are based on the DLL4/JAG1-Notch-VEGF signalling model (2) by assuming that there is no DLL4-JAG1 heterodimerization in the cells. In this scenario, the variation of JAG1 production rate does not affect the bifurcation diagram driven by DLL4 production rate.
(TIFF)

**S8 Fig. Differential affinities of JAG1-DLL4 and DLL4-DLL4 interactions.** (A) HIS epitope tagged DLL4 immunopurified from 293T cells using nickel-NTA agarose was incubated with the indicated combinations of (cell-expressed) HA epitope tagged DLL4 and JAG1. Protein Interactions were determined by Western blotting with the indicated antibodies. (B) Quantification of the bands shown in (A).
(EPS)

**S9 Fig. A DLL4-JAG1 heterodimer blocks DLL4-mediated Notch2 and Notch3 receptor cis inhibition and enables efficient Notch2 and Notch3 receptor transactivation.** (A) Schematic representation of luciferase reporter assays performed as described in the Methods. (B) 293T cells co-expressing a Notch activity luciferase reporter together with the indicated Notch2 receptor and ligand constructs (stalk cells) were co-cultured with cells stably expressing DLL4 (tip cells) to enable transactivation. DLL4-Δ(EGF7) and DLL4-Δ(EGF8) lack the EGF-like repeat 7 and the EGF-like repeat 8 (see Fig 3C), respectively. For each analysis, reporter activity was normalized using Renilla luciferase. Levels of ectopically expressed proteins were determined by Western blotting of cell lysates. Each condition in each experiment was performed in triplicate and error bars represent the standard deviation of the mean. Experiments were performed three times. Representative experiments are shown. (C) 293T

cells co-expressing a Notch activity luciferase reporter together with the indicated Notch3 receptor and ligand constructs (stalk cells) were co-cultured with cells stably expressing DLL4 (tip cells) to enable transactivation. DLL4-Δ(EGF7) and DLL4-Δ(EGF8) lack the EGF-like repeat 7 and the EGF-like repeat 8 (see Fig 3C), respectively. For each analysis, reporter activity was normalized using Renilla luciferase. Levels of ectopically expressed proteins were determined by Western blotting of cell lysates. Each condition in each experiment was performed in triplicate and error bars represent the standard deviation of the mean. Experiments were performed three times. Representative experiments are shown.
(EPS)

**S10 Fig. Protein levels of Notch receptors and ligands were determined by Western blotting of lysates corresponding to the following luciferase reporter assays: (A)** Fig 1A**. (B)** Fig 2A**. (C)** Fig 2E**. (D)** Fig 3B**. (E)** Fig 3E**.**
(EPS)

**S1 Video. Multicellular simulations of tip/stalk cell patterning in sprouting angiogenesis.** This video corresponds to Fig 5A with DLL4-JAG1 heterodimerization.
(GIF)

**S2 Video. Multicellular simulations of tip/stalk cell patterning in sprouting angiogenesis.** This video corresponds to Fig 5B without DLL4-JAG1 heterodimerization.
(GIF)

## Acknowledgments

We are very grateful to Maarten van Dinther and Lin-Ting Wu for excellent technical assistance with the experimental work. We thank members of the Department of Cell & Chemical Biology at the LUMC for helpful discussions and support, in particular, Prof. Peter ten Dijke, Dr Manuel Goncalves and Shaima Abdalla.

## Author contributions

**Conceptualization:** David A Baker.

**Data curation:** Daipeng Chen, Xinxin Liu, Haijiang Wang, David A Baker.

**Formal analysis:** Daipeng Chen.

**Investigation:** Daipeng Chen, Xinxin Liu, Haijiang Wang.

**Methodology:** Daipeng Chen.

**Project administration:** David A Baker.

**Supervision:** Roeland MH Merks.

**Writing – original draft:** David A Baker.

**Writing – review & editing:** Daipeng Chen, Xinxin Liu, Haijiang Wang, Roeland MH Merks.

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
