## [Decision Letter · Decision Letter 0]

12 Nov 2024

PCOMPBIOL-D-24-01720A new model of Notch signalling: evidence of a role for Notch ligand heterodimerizationPLOS Computational Biology Dear Dr. Baker, Thank you for submitting your manuscript to PLOS Computational Biology. After careful consideration, we feel that it has merit but does not fully meet PLOS Computational Biology's publication criteria as it currently stands. Therefore, we invite you to submit a revised version of the manuscript that addresses the points raised during the review process. Please note that one reviewer has indicated that the simulation code used in this study should be made available upon publication to comply with PLOS Data policy  and we ask you to please address this point in addition to those raised in the critiques. Please submit your revised manuscript within 30 days Jan 12 2025 11:59PM. If you will need more time than this to complete your revisions, please reply to this message or contact the journal office at ploscompbiol@plos.org. Please include the following items when submitting your revised manuscript:* A rebuttal letter that responds to each point raised by the editor and reviewer(s). You should upload this letter as a separate file labeled 'Response to Reviewers'. This file does not need to include responses to formatting updates and technical items listed in the 'Journal Requirements' section below.* A marked-up copy of your manuscript that highlights changes made to the original version. You should upload this as a separate file labeled 'Revised Manuscript with Track Changes'.* An unmarked version of your revised paper without tracked changes. You should upload this as a separate file labeled 'Manuscript'. If you would like to make changes to your financial disclosure, competing interests statement, or data availability statement, please make these updates within the submission form at the time of resubmission. Guidelines for resubmitting your figure files are available below the reviewer comments at the end of this letter. We look forward to receiving your revised manuscript. Kind regards, James R. FaederAcademic EditorPLOS Computational Biology Jason HaughSection EditorPLOS Computational Biology

Feilim Mac Gabhann

Editor-in-Chief

PLOS Computational Biology

Jason Papin

Editor-in-Chief

PLOS Computational Biology

**Journal Requirements:****Reviewers' comments:** Reviewer's Responses to Questions

**Comments to the Authors:**

Reviewer #1: In this manuscript, the authors develop a quantitative model for the Notch pathway that includes a novel effect of ligand heterodimerization. Through elegant experiments, they demonstrate a combinatorial effect of JAG1 and DLL1, two main ligands of the Notch pathway. They use this insight to develop a detailed biophysical model for Notch response. This work tackles the general question of combinatorial effects in signaling pathways, which is critical for understanding how cells process information from their environment. Impressively, by combining new experimental observation with quantitative modeling, the authors were able to identify an important feature of such combinatorial interactions, which has the potential to explain the differentiation of endothelial cells during angiogenic sprouting.

My main concern is about assumptions in the model that should be discussed more carefully. Specifically:

1. The authors study experimentally the capacity of DLL4, JAG1 and JAG2 to trans-activate and cis-inhibit Notch1. They show that JAG1 does not promote trans activation or cis inhibition. They then explain the first observation by lack of binding and the second observation by lack of dimer formation. Can the lack of binding itself be enough to explain both observations? While they compare the cis behavior to that of JAG2, there is no comparison of the trans behavior.

2. In the model, the author uses a single degradation parameter for all ligand monomers, dimers, and the Notch receptor. What is the motivation behind this assumption? How valid is it? Can this affect the results?

3. The model nicely describes the binding of ligands and receptors. However, it completely ignores possible reverse reactions through which complexes revert to their monomeric components. Critically, this allows the authors to solve the first two equations in (1) for D and J independently of the number of complexes or receptors. What is the justification for this assumption, and how would the result change if unbinding were allowed to occur?

Minor points:

4. While all the information is in the manuscript, writing more information within the figures could ease the reading.

- The immunoprecipitation panels could be more readable by explicitly stating the antibodies used for precipitation and staining.

- The legends show the Notch receptor as a single protein, but on the cells, two receptors bind to a single ligand. This is confusing, as the model considers a 1-1 binding between a single receptor and a single ligand.

- The cartoons for the sprouting angiogenesis simulations could gain from another panel of the in-vivo expected result for comparison.

5. Line 251: there might be a missing square root. The condition for B2 to be a small number is k_he >> \sqrt(k_ho).

Reviewer #2: The manuscript presents a development of an earlier study in PLOS Comp Bio from the same group. It provides a nice combination of cell-based assays and directed mathematical / computational modelling to explore the potential importance of homo- and hetero-dimerisation of Notch ligands in regulating the pattern of Notch signalling activity. By coupling the model of Notch ligand interactions with VEGF signalling, the model can account for the broad features of the initiation of sprouting angiogenesis.

Everything seems ok in the analysis and results. I feel that the use of "A new model of Notch signalling", as in the previous paper, is a bit of an overstatement. The effective mutual exclusion of the two unbound ligands is pretty much the same mechanism as the mutual exclusion of Notch and Delta in Sprinzak's cis inhibition model (2010, 2011), so the current results just amplify the idea that when molecules can bind into a variety of complexes, some of which are active and some inactive, then ultrasensitivity can result. I also feel that calling one's own previous recent paper "seminal" is a bit much...

Specific comments:

1. The authors should provide a bit more information about the simulations. For Fig 5A,B, where exactly is y = y_0 (used in the exponential decay function)? It is not clear. Also, how exactly does the coordinate y relate to the cell rows? It is not clear in the manuscript. For the simulations, what were the initial and boundary conditions used?

2. In Fig. 5A,B, it isn't clear to me why the authors chose conditions that restrict Notch activation to the first row of cells, so we have simulations where nothing much is happening in most of the cells. What happens if the VEGF gradient is shallower / longer-range? As it is, the model shows a lateral inhibition-like pattern in row 1 (sometimes also row 2); would this emerge in more rows of cells for a shallower VEGF gradient? It is also not clear how that pattern of Notch signalling activity translates to the biology of sprouting angiogenesis. I understand that this corresponds to the selection of tip cells, but is the period-2 spacing appropriate for the real system? Is the initial pattern of Notch activity period-2 in reality, or is it longer-range (as is typically the case? A bit more information to link the simulations to the target system would be helpful.

3. Fig. 4 shows slices through the (b_D, b_J) plane. Would it be possible to present a full picture of NICD across the plane (a 2D plot)? I think I know what it will look like, but it would be good to see how NICD depends on the full range of production rates.

4. In this model, as with the Sprinzak model of cis- and trans- binding, binding is assumed to be irreversible. So Notch and its ligands form cis- complexes that never dissociate. I think it would be good to state this assumption. I presume those complexes are going somewhere. Should I assume they are degraded / endocytosed?

5. Equation (6) is really an approximation (the authors use the expression "reduces to", but that would strictly only be in the limit). I think it should be written as such.

6. On line 343, I think the statement that the model "correctly predicts" is an overstatement; that relationship between VEGFR2 activity and high DLL4 expression is put in the model (the production rate of DLL4 is an increasing Hill function of VEGFR2 activity...)

7. The use of "expression" on line 346 is unhelpfully vague. I can't work out what the authors mean here. I think expression refers to the steady state amount of free ligands in the plasma membrane, since the production rates of ligands (what one might expect to be meant by "expression") are spatially uniform. This may seem a bit semantic, but I think it is important to be clear here, since it is at the heart of the proposed patterning mechanism.

8. A couple of minor typos: On line 455, "principle" should be "principal". In Table 1, "typic" should be "typical"

Reviewer #3: The work tackles an important question regarding cis-inhibition and is valuable for the field. The biochemical approach to studying cis-inhibition is simple but effective. The computational model nicely integrates the findings and makes interesting predictions. However, some technical and conceptual issues need to be addressed to support some of the claims and to improve the readability.

1) At which cell confluency was the cis-inhibition experiments performed? The authors should at least briefly explain whether they would expect co-expressing ligands in the “receiver” cells to increase the transactivation between two adjacent “receiver” cells and whether they considered the cell confluency as a factor that can convolute the experimental results.

2) In general, figures with Western blot images need further clarification and addition of the necessary controls. Please see below:

In Figure 1D, the size of all three Notch paralogs seems to be the same, even though the Notch3 receptor should be about 20kD smaller. The authors should explain what this band corresponds to and perhaps show the full membrane together with the protein ladder in the supplementary data. The size of the bands, and blots of an appropriate housekeeping protein should be added. Additionally, the blots are mis-aligned and do not match the sample labels properly.

While how the receptor and the ligand constructs were tagged were explained in the methods section, it is difficult to understand the construct identity this by looking at the figure, since only the tag names are shown. Adding this information to the figures either in the cartoons or on the blots is recommended.

Figure 2: The blots from the immunoprecipitation experiments are difficult to understand. The authors should annotate the blots with the information which proteins were pulled down and what they were probed for afterwards (e.g. IP:His, WB: His and IP:His, WB:HA) to make them easier for the reader to understand.

Figure 5S: Considering the Notch1 construct was tagged in the C-terminal, wouldn´t the authors expect to see the cleavage products of the Notch receptor in their blots when it is activated by the ligand-expressing cells? Judging by the Dll4, Jagged1, Notch1 co-blots the authors are only showing the full length Notch1 precursor. Additionally, the authors should again annotate the protein size on these blots, and possibly blot the same samples for the cleaved NICD to further support the reporter assay results if it is not possible to visualize the cleavage products otherwise.

3) Line 832, 836, 851, 854: “tissue-culture cells” sounds off. Please revise.

4) There seems to be major issues relating to the coherence of the manuscript. For example, the use of the Notch3 ECD protein and of DAPT as a control condition for the luciferase assays were elusive in the text and in the figures, even though they were mentioned in the material and methods section(?). The authors must make sure that the writing is consistent with the figures.

5) For all figures, the appropriate statistic tests should be done for the experimental data, and corresponding graphs should be annotated with the results.

Further deficiencies:

• Figures need to be organized better. In some cases, they lack necessary information.

• Issues with consistency between the text and the data.

The manuscript requires

• Additional experiments

• Additional information

• Revision of the text

**Have the authors made all data and (if applicable) computational code underlying the findings in their manuscript fully available?**

Reviewer #1: Yes

Reviewer #2: **No: ** I can't see any information on availability of the code used for simulations.

Reviewer #3: Yes

PLOS authors have the option to publish the peer review history of their article (what does this mean? ). If published, this will include your full peer review and any attached files.

**Do you want your identity to be public for this peer review?** For information about this choice, including consent withdrawal, please see our Privacy Policy .

Reviewer #1: No

Reviewer #2: No

Reviewer #3: No

---

## [Decision Letter · Decision Letter 1]

25 Jan 2025

Dear Dr Baker,

We are pleased to inform you that your manuscript 'A model of Notch signalling control of angiogenesis: evidence of a role for Notch ligand heterodimerization' has been provisionally accepted for publication in PLOS Computational Biology.

Best regards,

James R Faeder

Academic Editor

PLOS Computational Biology

Jason Haugh

Section Editor

PLOS Computational Biology

Reviewer's Responses to Questions

**Comments to the Authors:**

Reviewer #1: The authors have adequately addressed my concerns. The paper addresses an important question in a rigorous manner, and should be published.

My only comment is a small typo. In Line 149, I believe that the phrase “(and JAG2)” seems out of place.

Reviewer #2: The authors have revised the manuscript in a way that addresses all the points I made in the first round of review. I am therefore happy to recommend acceptance for publication.

Reviewer #3: The authors have appropriately addressed the reviewers comments and I am happy to accept the manuscript for publication.

**Have the authors made all data and (if applicable) computational code underlying the findings in their manuscript fully available?**

Reviewer #1: None

Reviewer #2: Yes

Reviewer #3: Yes

PLOS authors have the option to publish the peer review history of their article (what does this mean? ). If published, this will include your full peer review and any attached files.

**Do you want your identity to be public for this peer review?** For information about this choice, including consent withdrawal, please see our Privacy Policy .

Reviewer #1: No

Reviewer #2: No

Reviewer #3: No

---

## [Editor Report · Acceptance letter]

PCOMPBIOL-D-24-01720R1

A model of Notch signalling control of angiogenesis: evidence of a role for Notch ligand heterodimerization

Dear Dr Baker,

I am pleased to inform you that your manuscript has been formally accepted for publication in PLOS Computational Biology. Your manuscript is now with our production department and you will be notified of the publication date in due course.

With kind regards,

Zsofia Freund
